# Hybrid Query Strategy with Diversity–Weighted Metropolis–Adjusted Langevin Algorithm

## Abstract

Although deep learning has achieved remarkable success in various fields, most of these advances typically rely on a large-scale well-annotated dataset. However, in real-world applications, collecting labeled data is often expensive and time-consuming. Active Learning (AL) has emerged as a promising solution to mitigate labeling costs by selectively querying the most informative instances for annotation. In particular, hybrid AL methods have been gaining attention by integrating multiple acquisition criteria such as uncertainty, diversity, or representativeness as a joint function. In this paper, we propose a novel hybrid active learning method named Diversity-Weighted Metropolis-Adjusted Langevin Algorithm (DW-MALA). Our method precisely approximates the data distribution by leveraging gradient-based Langevin dynamics, and selects instances from high-density regions using a representativeness score derived from density estimates. Simultaneously, a diversity score is incorporated by measuring the distance to the nearest labeled instance, which also ensures coverage of low-density regions. The quantitative and qualitative analyses demonstrate the effectiveness of DW-MALA in selecting diverse and representative samples under a limited labeling budget, compared to the baselines.

## 1 Introduction

The recent demand for deep learning is increasing across a wide range of domains such as computer vision, natural language processing, signal processing, etc. Most of these advances have been achieved under a large-scale dataset that is annotated the groundtruth label for the task. In real-world scenarios, however, acquiring such labeled data is often prohibitively expensive and time-consuming, as it requires extensive manual effort from a human annotator, or an oracle. Moreover, the scalability of manual annotation becomes a major bottleneck as data volume increases. To address these challenges, **active learning** (Settles, 2009) has emerged as a compelling paradigm by querying the most informative instances under a limited annotation budget. The goal of active learning is to prioritize and label samples that will most benefit the model, enabling progressive enhancement through iterative learning.

Existing active learning methods can be broadly categorized into uncertainty-based, diversity-based, representativeness-based, and hybrid methods, by differentiating the definition of informativeness. *Uncertainty-based methods* select instances for which the model is most uncertain, typically measured by entropy in prediction probabilities (Settles, 2009; Houlsby et al., 2011) or metrics estimated by auxiliary networks (Yoo & Kweon, 2019; Sinha et al., 2019). *Diversity-based methods* aim to select instances that are well spread out in the feature space, with the goal of covering underexplored areas and reducing redundancy of the input space (Sener & Savarese, 2017). *Representativeness-based methods* prioritize instances that best reflect the overall data distribution, typically by leveraging clustering or density estimation techniques (Wang & Ye, 2015). Recently, *hybrid methods* have gained attention to integrate multiple acquisition strategies—such as uncertainty, representativeness, and diversity—into a unified framework to leverage their complementary strengths (Ash et al., 2020). By jointly considering different criteria, hybrid methods can achieve more effective selection in complex learning scenarios.

In this paper, we focus on a hybrid method, where we integrate diversity with representativeness. Specifically, it is essential to model the underlying data distribution to quantify representativeness, while most of the previous methods have focused on designing heuristic metrics. For example,

Density-Weighted Diversity Strategy (DWDS) (Wang et al., 2021) utilizes cosine similarity as the metric, assuming that samples at the high-density region shows a high value of the averaged cosine similarity between neighbors. Also, Maximum Mean Discrepancy (Gretton et al., 2012) between labeled and unlabeled dataset has been adopted as a minimization objective, so that the selected instances cover the overall data distribution (Wang & Ye, 2015). Despite their empirical effectiveness, the previous methods either depend on manually crafted measures or indirectly simulate the data distribution, which may fail to capture the full complexity of the underlying distribution precisely.

To address these limitations, we propose a novel hybrid active learning method, termed **Diversity-Weighted Metropolis-Adjusted Langevin Algorithm**, or DW-MALA. DW-MALA explicitly models the data distribution via gradient-based Langevin dynamics to compute a representativeness score, facilitating global exploitation of the data manifold. In parallel, a diversity score is incorporated to encourage local exploration, ensuring that the selected instances are not only representative but also diversely dispersed across the feature space. Specifically, we adopt the Metropolis-Adjusted Langevin Algorithm (MALA), which is a Markov Chain Monte Carlo (MCMC) sampling algorithm based on the Langevin Stochastic Differential Equation (SDE), to estimate representativeness. MALA generates a proposal sample with gradient-based information and determines whether to accept each proposal based on the Metropolis-Hastings criterion. The collected proposals by MALA are then used to approximate the entire unlabeled data distribution. For each unlabeled instance, the estimated density

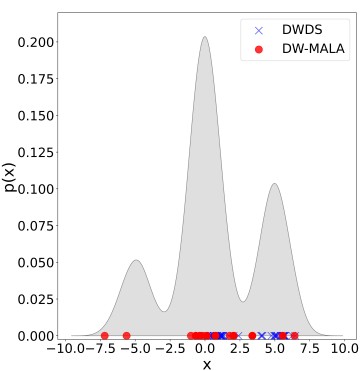

Figure 1: Toy Example for Comparing DWDS and DW-MALA on Gaussian Mixture Distribution

value is employed as the representativeness score of the instance itself. Furthermore, we incorporate a diversity score to guide the acquisition process toward underexplored or minor modes. To quantify diversity, we compute the minimum distance between each unlabeled instance and its nearest labeled instance in the feature space. By integrating both the representativeness and diversity scores into a unified acquisition function, we propose a novel active learning algorithm that:

- Accurately models the underlying data distribution.

- Exploits high-density regions to acquire representative instances.

- Explores low-density regions to prevent overfitting to dominant modes and ensure sample diversity.

The effective balance between exploration and exploitation is illustrated in Figure 1, where we simulate a toy example on a 1-dimensional Gaussian mixture distribution. In the figure, the baseline method DWDS, visualized as blue crosses, fails to select instances in the leftmost minor mode, indicating limited coverage of the data space. In contrast, the instances selected by DW-MALA, visualized as red dots, are distributed broadly along the x-axis, successfully covering both high- and low-density regions. These results highlight that DW-MALA not only exploits the entire data space by considering representativeness but also explores the minor mode by incorporating diversity into the acquisition function. To the best of our knowledge, this is the first work to adopt Langevin SDE into the design of an active learning algorithm.

## 2 RELATED WORK

### 2.1 ACTIVE LEARNING

Active learning (Settles, 2009) is a paradigm in machine learning that aims to improve model performance with minimal labeling cost by selectively acquiring the most informative data instances. Given the unlabeled dataset $\mathcal{D}_U$, an acquisition function identifies a subset $\mathcal{D}_Q$ by ranking instances according to an informativeness score and selecting the top-$K$ instances. The selected query set is then annotated and added to the labeled dataset $\mathcal{D}_L$, which is subsequently used to update the task model. Among various query strategies, uncertainty-based methods prioritize instances for which the current model exhibits high predictive uncertainty, which include Entropy (Shannon, 1948), Variation Ratio (Freeman, 1965), predicted loss (Yoo & Kweon, 2019), discrimination from labeled dataset (Sinha et al., 2019), etc. These methods rely on model prediction to identify ambiguous instances near decision boundaries. Diversity-based methods aim to enhance the coverage of the input space by

selecting heterogeneous instances, thereby mitigating redundancy and reducing the risk of overfitting to a narrow subset of the labeled set. Typical implementations include k-Means clustering (Har-Peled & Kushal, 2005; Lin et al., 2024) or linear programming (Sener & Savarese, 2017) to construct a diverse acquisition set. Finally, hybrid method have been proposed to overcome the limitations of individual acquisition criteria and aggregate the strengths. For example, the prior work of (Ash et al., 2020) uses gradient embedding to measure the uncertainty, and applies the k-Means++ seeding algorithm to select a diverse set of uncertain samples.

## 2.2 Representativeness-based Active Learning

Since this paper proposes a novel hybrid-based active learning method that integrates both diversity and representativeness, we provide a detailed overview of the most relevant prior methods within this domain. In the prior work of (Wang & Ye, 2015), the acquisition function selects a new query set by jointly minimizing empirical loss and distributional mismatch. Specifically, the latter objective, which is in charge of representativeness, is enforced via a Maximum Mean Discrepancy between the feature embedding of the entire unlabeled dataset and that of the query set. By minimizing the difference, the strategy forces the empirical distribution of labeled samples to match that of the full dataset, thereby ensuring that the chosen query set is maximally representative of the entire unlabeled set. Another notable prior work of (Wang et al., 2021) formulates its query strategy to simultaneously promote diversity and representativeness. In this work, diversity is encouraged by selecting instances that are maximally dissimilar from the current labeled set, while representativeness is promoted by favoring instances residing in high-density regions of the unlabeled pool. To estimate representativeness in specific, the method measures the cosine similarity with the nearest labeled instance under the assumption that samples in dense regions exhibit high similarity to other data points, thereby serving as good representatives of the overall distribution. However, prior works rely on feature-based heuristics or indirect approximations that fail to accurately capture the underlying data distribution, making it difficult to ensure that the selected instances are truly representative.

## 2.3 Metropolis–Hastings Algorithm

The Metropolis–Hastings (MH) algorithm (Hastings, 1970) is a special case of Markov Chain Monte Carlo (MCMC) (Metropolis et al., 1953) method that generates samples from a desired target distribution $\pi(x)$ by constructing a reversible Markov chain whose stationary distribution equals $\pi$. Given the current state $x_t$, a candidate state $x'$ is sampled from a proposal density $q(x' \mid x_t)$, and is accepted with probability of:

$$\alpha(x_t, x') = \min\left(1, \frac{\pi(x')q(x_t|x')}{\pi(x_t)q(x'|x_t)}\right). \tag{1}$$

To decide whether to accept $x'$, a uniform random number, $r$, is drawn from the interval $[0, 1]$. The next state of the Markov chain is then determined as:

$$x_{t+1} = \begin{cases} x', & \text{if } r \leq \alpha(x_t, x') \\ x_t, & \text{otherwise.} \end{cases} \tag{2}$$

This acceptance rule ensures that the resulting Markov chain satisfies detailed balance, thereby guaranteeing convergence to the target distribution $\pi(x)$.

# 3 Methodology

## 3.1 Langevin Stochastic Differential Equation

Our proposed method, DW-MALA, is built upon Langevin dynamics (Langevin et al., 1908), which describes the motion of a particle under the influence of both deterministic and stochastic forces. For a particle of mass $m$, the Langevin equation is given by:

$$m\frac{d^2x_t}{dt^2} = -\gamma\frac{dx_t}{dt} - \nabla U(x_t) + \sqrt{2}\frac{dW_t}{dt}, \tag{3}$$

where $x_t$ is the position of the particle at time $t$; $\gamma$ is the friction coefficient; $U(x_t)$ is the potential function whose gradient represents the deterministic force; and $W_t$ denotes a Wiener process (i.e., standard Brownian motion) whose gradient results in a stochastic force term, $\sqrt{2}\frac{dW_t}{dt}$.

When it comes to a problem of probability density estimation for active learning, we treat each data instance as a particle and hence assume that $m = 0$, which simplifies the Langevin equation to:

$$\frac{dx_t}{dt} = -\frac{1}{\gamma}\nabla U(x_t) + \frac{\sqrt{2}}{\gamma}\frac{dW_t}{dt}. \tag{4}$$

Since solving Eq.(4) is a complicated problem, we assume that $U(x) = \frac{1}{2}kx^2$ for an arbitrary constant $k$, and represent Eq.(4) as:

$$\frac{dx_t}{dt} = -\frac{k}{\gamma}x_t + \frac{\sqrt{2}}{\gamma}\frac{dW_t}{dt}. \tag{5}$$

By introducing $u = -\frac{k}{\gamma}$ and $\sigma = \frac{\sqrt{2}}{\gamma}$, we then get the simplified version as:

$$dx_t = ux_tdt + \sigma dB_t, \tag{6}$$

where we represent the derivative of the Wiener process, $\frac{dW_t}{dt}$, as $dB_t$. Then, we get the solution for Eq.(6) as below with its detailed steps and assumptions provided in Appendix A.

$$x_t = x_0e^{ut} + \sigma e^{ut}\int_0^t e^{-us}dB_s, \tag{7}$$

Here, $x_0$ is a randomly initialized data point used to begin the distribution estimation process. Since the solution of Eq.(7) involves a stochastic integral, we discretize the process using the Euler-Maruyama method (Maruyama, 1955). For a time step $\Delta t$, the discretized approximation is given by:

$$x_{t_{n+1}} = x_{t_n} + ux_{t_n}\Delta t + \sigma\Delta B_{t_n}, \tag{8}$$

where $\Delta B_{t_n} \sim \mathcal{N}(0, \Delta t)$. Here, the continuous time variable $x_t$ is now represented in discrete time as $x_{t_n}$. Finally, to generalize this formulation, we substitute the definitions of $u$, $\sigma$, and $\nabla U(x) = kx$ into Eq.(8), yielding:

$$x_{t_{n+1}} = x_{t_n} - \frac{1}{\gamma}\nabla U\left(x_{t_n}\right)\Delta t + \frac{\sqrt{2\,\Delta t}}{\gamma}Z_n, \tag{9}$$

where $Z_n \sim \mathcal{N}(0, 1)$.

### 3.2 METROPOLIS-ADJUSTED LANGEVIN ALGORITHM

In high-dimensional settings where efficient sampling is essential, the Metropolis-Adjusted Langevin Algorithm (MALA) improves upon the traditional Metropolis-Hastings (MH) algorithm by leveraging gradient information from the target distribution to guide the proposal generation more effectively.

For the purpose of sampling from a probability distribution, we assume that the target distribution $\pi(x)$ follows a Boltzmann distribution (Leimkuhler et al., 2016):

$$\pi(x) \propto \exp\left(-\beta\,U(x)\right). \tag{10}$$

To apply the solution of Eq.(9), we follow prior work (Roberts & Tweedie, 1996) and set $\beta = 1$ and $\gamma = 1$. While this simplification may lead to suboptimal convergence in certain cases, we can benefit from practical merits by reducing computational complexity. By setting $\Delta t = \frac{\epsilon^2}{2}$, MALA samples a candidate state $x'$ from the current state $x_t$ using gradient information as follows:

$$x' = x_t + \frac{\epsilon^2}{2}\nabla\log\pi(x_t) + \epsilon Z, \tag{11}$$

where $Z \sim N(0, I)$ denotes standard Gaussian noise, and $\epsilon$ is the step size. The term $\nabla\log\pi(x_t)$, i.e., the gradient of the log-target distribution, guides the sampler toward regions of higher probability density, thereby improving sampling efficiency.

The proposed state $x'$ is accepted with the following acceptance probability:

$$\alpha(x_t, x') = \min\left(1, \frac{\pi(x')q(x_t|x')}{\pi(x_t)q(x'|x_t)}\right), \tag{12}$$

where we set the proposal distribution, $q(x'|x_t)$, as a Gaussian mixture distribution so that we can capture multiple modes around the current state. If $x'$ is accepted, we set $x_{t+1} = x'$; otherwise, we maintain the current state as $x_{t+1} = x_t$. It should be noted that by incorporating the gradient of the target distribution, MALA generates more informed proposals than MH, making it particularly effective in high-dimensional settings where traditional random-walk-based methods often struggle.

### 3.3 KERNEL DENSITY ESTIMATION

For a novel active learning method, our goal is to construct the acquisition function that quantifies the representativeness of each unlabeled instance. To this end, we adopt the Kernel Density Estimation (KDE) (Silverman, 2018), which is a nonparametric estimator of the probability density function $\pi(x)$, based on a set of samples $\mathcal{S}_{mala} = \{x_i\}_{i=1}^M$ generated by MALA sampler. To start with, we represent the estimated density function $\hat{\pi}(x)$ using a Dirac-Delta function $\delta(\cdot)$ as:

$$\hat{\pi}(x) = \frac{1}{M} \sum_{i=1}^{M} \delta(x - x_i), \tag{13}$$

$$\text{where} \quad \delta(x - x_i) = \begin{cases} 0, & \text{if } x \neq x_i \\ \infty, & \text{if } x = x_i \end{cases}, \quad \int_{-\infty}^{\infty} \delta(x - x_i)\, dx = 1. \tag{14}$$

Then, KDE approximates $\hat{\pi}(x)$ using a Kernel function $\Phi(\cdot)$ as:

$$\hat{\pi}_{KDE}(x) = \frac{1}{Mh} \sum_{i=1}^{M} \Phi\big(x - x_i\big), \tag{15}$$

where $h$ controls the width of the Dirac-Delta function for the samples.

Finally, we estimate the density for each unlabeled instance $x_u \in \mathcal{D}_U$, and define the representativeness score of $x_u$ as:

$$\mathcal{A}_{rep}(x_u) = \hat{\pi}_{KDE}(x_u), \tag{16}$$

which is the KDE-based density estimate. It is worth noting that this score prioritizes instances located in high-density regions, effectively guiding the acquisition toward representative areas of the data distribution and thereby promoting *exploitation*.

### 3.4 DIVERSITY-WEIGHTED MALA FOR ACTIVE LEARNING

In addition to the representativeness score for exploitation, we also incorporate the diversity score into the joint acquisition function to encourage exploration. Specifically, for each unlabeled instance $x_u \in \mathcal{D}_U$, we calculate the diversity score $\mathcal{A}_{div}(x_u)$ as the minimum distance to the labeled instance $x_l \in \mathcal{D}_L$ as below, where we use Euclidean distance between feature maps extracted from the penultimate layer of the task model:

$$\mathcal{A}_{div}(x_u) = \min_{x_l \in \mathcal{D}_L} \text{Dist}(x_u, x_l). \tag{17}$$

This score quantifies *exploration* by favoring points that are distant from existing labeled instances. Having said that, we can cover minor modes of the data distribution that may otherwise be overlooked by representativeness-based acquisition alone. Finally, we define the joint acquisition score of our DW-MALA, by combining the representativeness and diversity scores as:

$$\mathcal{A}_{DWMALA}(x_u) = \mathcal{A}_{div}(x_u)^{\alpha} \cdot \mathcal{A}_{rep}(x_u)^{1-\alpha}, \tag{18}$$

where $\alpha \in [0, 1]$ controls the trade-off between exploitation of high-density regions and exploration of low-density regions. At each acquisition iteration, the top-$K$ instances with the highest joint scores are selected from $\mathcal{D}_U$, annotated by the oracle, and added to the labeled dataset $\mathcal{D}_L$. The task model is then retrained on the updated $\mathcal{D}_L$, which is repeated for $N$ iterations until the labeling budget is exhausted. The complete procedure is outlined in Algorithm 1 of Appendix B.

In summary, our method achieves a balanced acquisition strategy by combining exploration and exploitation; the distance-based term encourages selection from underexplored low-density regions, while the density-based term focuses on representative high-density regions. It should be noted that the proposed version of DW-MALA, however, is not the only available format, and the robustness of DW-MALA is demonstrated through various experiments, such as different distance metrics for similarity and different combinations for the hybrid strategy, as provided in Appendix C.

## 4 EXPERIMENT

### 4.1 IMAGE CLASSIFICATION

#### 4.1.1 SETTINGS AND BASELINES

We conducted image classification on three benchmarks including CIFAR-10, CIFAR-100 (Krizhevsky et al., 2009), and Tiny-ImageNet (Le & Yang, 2015); and two imbalanced dataset

based on CIFAR-10. We used VGG16 (Simonyan & Zisserman, 2014) for the task model and trained with RMSProp (Tieleman, 2012) optimizer with a learning rate of 1e-3 and a batch size of 32. Each dataset uses a different acquisition schedule: CIFAR-10 starts from 1,000 labeled instances and selects 1,000 instances for 9 additional acquisition iterations each; CIFAR-100 selects 5,000 instances for 5 additional acquisition iterations starting from 5,000 labeled instances; and Tiny-ImageNet starts with 20,000 labeled instances and acquires 20,000 instances per acquisition iteration for 4 additional rounds. After each acquisition, the model is re-initialized and trained for 50, 200, and 350 epochs, for each dataset respectively.

For DW-MALA, we set $M = 20,000$, $h = 1$, and a Gaussian kernel $\Phi$ in Eq.(15); which are consistently applied across all tasks. For evaluation, we compare DW-MALA against several baselines: including Random, Entropy (Shannon, 1948), Coreset (Sener & Savarese, 2017), Variational Adversarial Active Learning (VAAL) (Sinha et al., 2019), Learning Loss for Active Learning (LL4AL) (Yoo & Kweon, 2019), Batch Active Learning by Diverse Gradient Embeddings (BADGE) (Ash et al., 2020), and ProbCover (Yehuda et al., 2022).

### 4.1.2 EXPERIMENT ON BENCHMARK DATASET

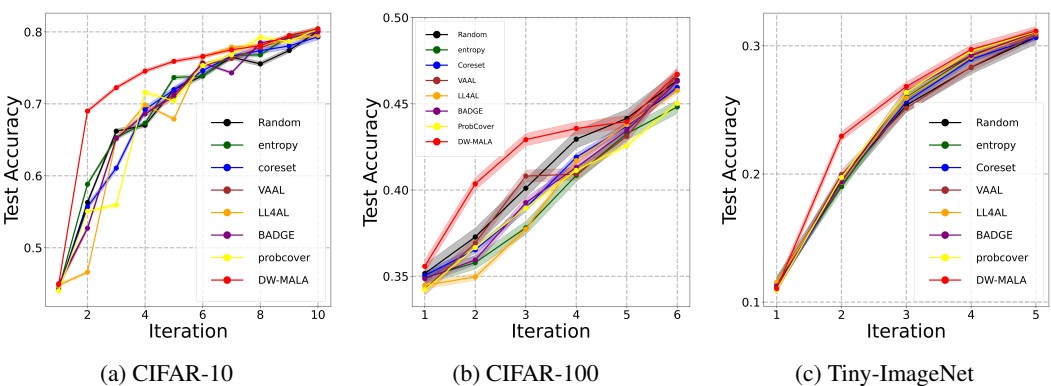

|  |  |  |
|:---:|:---:|:---:|
| (a) CIFAR-10 | (b) CIFAR-100 | (c) Tiny-ImageNet |

Figure 2: Benchmark Classification

The experimental results are summarized in Figure 2, where we repeated the experiments for five independent trials. On CIFAR-10, DW-MALA achieves a final accuracy of 80.41%, which is competitive with Entropy (80.45%) and outperforming Random (80.27%), Coreset (79.31%), VAAL (78.11%), LL4AL (79.52%), BADGE (80.01%), and ProbCover (80.39%). Notably, DW-MALA shows a substantial performance gain in the initial acquisition iterations, indicating its effectiveness in selecting informative samples under limited annotation budgets. The effect of DW-MALA is also

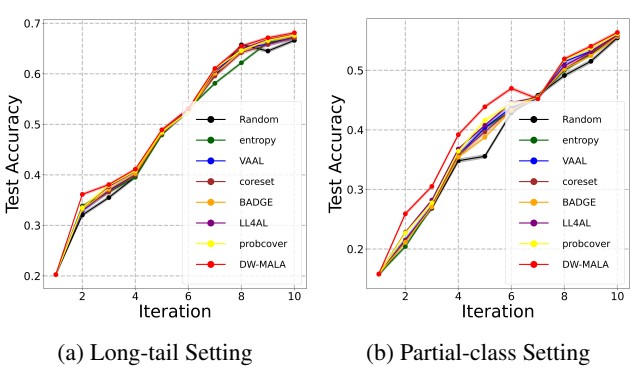

|  |  |
|:---:|:---:|
| (a) Long-tail Setting | (b) Partial-class Setting |

Figure 3: Imbalanced CIFAR-10 Classification

investigated in other complex benchmarks. On CIFAR-100, DW-MALA achieves the highest accuracy of 46.71%, which surpasses the baselines of Random (46.36%), Entropy (44.82%), Coreset (45.95%), VAAL (46.55%), LL4AL (45.77%), BADGE (46.23%), and ProbCover (45.03%). Similarly, DW-MALA outperforms baselines on Tiny-ImageNet with the final accuracy of 31.15%, where baselines reported Random (30.66%), Entropy (30.97%), Coreset (30.64%), VAAL (30.85%), LL4AL (31.01%), BADGE (31.05%), and ProbCover (31.09%). The gap between DW-MALA and baselines is again maximized at the initial acquisition iterations.

We attribute the advantage of DW-MALA, particularly in early iterations, to its ability to exploit representative data instances from various modes identified by the approximated data distribution of Eq.(15), while simultaneously encouraging diversity by explore samples that are distant from the

existing labeled set. This combination enables the early construction of a training set that is both representative and diverse, allowing the model to generalize more effectively to the overall data distribution and thus improving test performance.

### 4.1.3 EXPERIMENT ON IMBALANCED DATASET

To address the concern regarding the limited evaluation under class imbalance, we conducted additional experiments on CIFAR-10 by introducing two imbalanced settings, which are 1) long-tail setting where the number of data instances in each class decreases progressively with an imbalance factor of 0.1, and 2) partial-class setting where half of the classes contain all data instances while the remaining half use only 10% of instances. For both settings, we repeated the experiments for five independent trials. The results in Figure 3 show that DW-MALA consistently outperforms the baselines, especially showing a large gap in the early acquisition iterations. The results demonstrate that the diversity and representativeness scores indeed contribute to the improved data selection under imbalanced conditions.

### 4.1.4 TIME COMPLEXITY

We compare the wall-clock computation time required for a single acquisition iteration across all methods, which is reported in the Table 1 below. The results indicate that DW-MALA is competitive in terms of computational cost while achieving the highest performance across benchmarks.

| **Method** | Random | Entropy | CoreSet | VAAL | BADGE | LL4AL | ProbCover | DW-MALA |
|---|---|---|---|---|---|---|---|---|
| **Time (sec)** | 12.17 | 45.64 | 768.55 | 1003.52 | 946.92 | 938.72 | 472.72 | 884.96 |

Table 1: Wall-clock Time Comparison of Single Acquisition between DW-MALA and Baselines

While DW-MALA involves MALA-based sampling, its overall cost is comparable to or lower than other baseline methods such as BADGE, VAAL, and LL4AL. Notably, BADGE incurs high costs due to gradient embedding computation and k-Means++ seeding. Also, VAAL and LL4AL require training additional modules such as discriminator, VAE, or loss-prediction network. Despite similar computation time, however, DW-MALA consistently outperforms all baselines, particularly in early acquisition iterations where active learning provides the most benefit, as shown in Figures $2-3$. This indicates that DW-MALA offers a favorable trade-off between computational efficiency and query performance. It should be noted that the MALA sample size, $M$, affects the time complexity of the DW-MALA algorithm, i.e., time complexity grows with $M$; but the test accuracy remains stable with varying $M$ as shown in Figure 8 of Appendix D, indicating the robustness of DW-MALA.

## 4.2 QUALITATIVE ANALYSIS ON IMAGE CLASSIFICATION

### 4.2.1 ABLATION STUDY

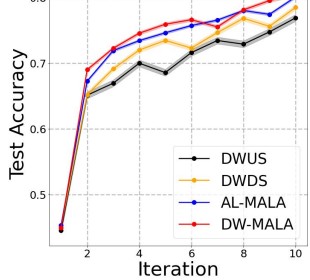

| Method | Uncertainty | Diversity | Representativeness |
|---|---|---|---|
| AL-MALA | X | X | $\Delta$ (MALA) |
| DWUS | O | X | O (cosine similarity) |
| DWDS | X | O | O (cosine similarity) |
| **DW-MALA** | X | O | O (MALA) |

Table 2: Model Description for Ablation Study

Figure 4: Ablation Study Results

DW-MALA achieves a balanced trade-off between exploration and exploitation. To further evaluate the robustness of DW-MALA, we conducted an ablation study on CIFAR-10 by controlling its core components. The compared variant models are described in Table 2. First, AL-MALA directly queries the instances generated by MALA; i.e., we only implement `line1-7` in Algorithm 1 and annotate $\mathcal{S}_{mala}$ to add to the labeled set. Second, Density-Weighted Diversity Sampling (DWDS) (Wang et al., 2021) combines diversity and representativeness, while representativeness differs from DW-MALA by using cosine similarity for `line9` in Algorithm 1. Third, Density-Weighted Uncertainty Sampling (DWUS) replaces the diversity term in DWDS with the uncertainty term measured by Entropy.

As shown in Figure 4, DW-MALA is the best performer with a final test accuracy of 80.40%; outperforming baselines of AL-MALA (80.08%), DWDS (78.46%), and DWUS (76.88%). We attribute the performance degradation of DWDS and DWUS to the use of cosine similarity as a metric for representativeness; that is, both models might fail to capture the data distribution precisely. AL-MALA, which benefits from Langevin-based density simulation, records the second-best performer. However, AL-MALA does not select top-$K$ instances with high density, but rather randomly selects according to the estimated density. Furthermore, AL-MALA also does not consider diversity for the selection, which leads to suboptimal performance. Finally, DW-MALA leverages density-driven exploitation and distance-based exploration, enabling it to construct a more representative and diverse query set; hence shows the most superior performance.

### 4.2.2 UMAP ANALYSIS

To provide additional evidence for the quantitative findings, we analyzed the data instances using Uniform Manifold Approximation and Projection (UMAP) (McInnes et al., 2018). In Figures 5a $-$ 5b, colored dots represent the labeled dataset with the corresponding label, gray dots represent the unlabeled dataset, and black triangles represent the selected dataset at the current acquisition iteration. As shown in Figure 5a, DWDS, which is our main baseline, selects instances that appear globally distributed to some extent, but still exhibit concentration in specific regions. We conjecture that this bias is due to the limited representation of the entire unlabeled data distribution when using an inaccurate measure, such as cosine similarity, for the representativeness score. In contrast, Figure 5b illustrates that DW-MALA selects instances more broadly across the whole data space, resulting in a globally distributed selection pattern. We provide additional UMAP visualization for all baselines across acquisition iterations in Appendix E.

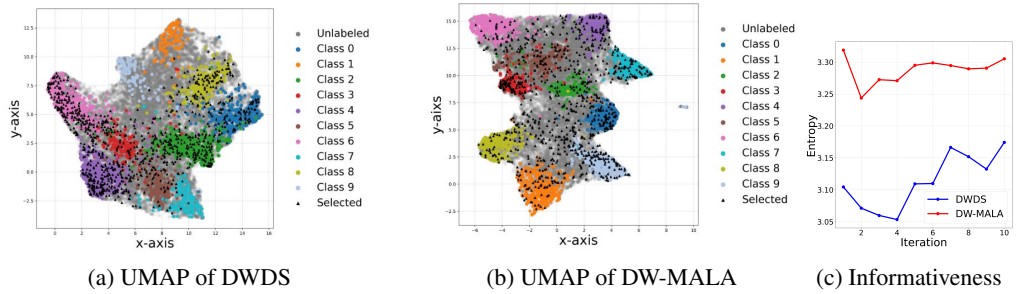

| (a) UMAP of DWDS | (b) UMAP of DW-MALA | (c) Informativeness |
|---|---|---|

Figure 5: Comparison of Selected Instances between DWDS and DW-MALA

The difference in selection behavior between DWDS and DW-MALA is further supported by Figure 5c, which reports the entropy of label distributions for selected instances across acquisition iterations. In the figure, DWDS exhibits consistently low entropy, indicating that its selected instances are label-imbalanced. In contrast, DW-MALA maintains higher entropy, suggesting that it acquires instances whose labels are evenly distributed. This indicates that DW-MALA performs sampling in a more globally distributed manner, which effectively covers both major and minor modes of the data distribution. Detailed label distributions at each acquisition iteration are provided in Appendix F.

### 4.2.3 SENSITIVITY ANALYSIS ON $\alpha$

In this analysis, we examine the robustness of DW-MALA with respect to the hyperparameter. Having said that, $\alpha$, which controls the balance between representatives and diversity in the joint acquisition score of Eq.(18), is the key hyperparameter that affects the performance. As shown in Figure 6, we varied $\alpha$ over $\{0, 0.25, 0.5, 0.75, 1\}$ and measured test accuracy at each acquisition iteration on CIFAR-10. The best performance was observed at $\alpha = 0.75$; but the differences across settings were marginal, suggesting that DW-MALA is robust to variations in $\alpha$. It is worth noting that when $\alpha = 0$ or $\alpha = 1$, the acquisition relies solely on either representativeness or diversity, reducing DW-MALA to a single-criterion method, which leads to suboptimal performance compared to hybrid configurations.

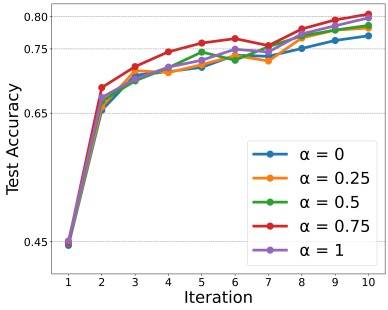

Figure 6: Sensitivity Analysis on $\alpha$

### 4.3 OBJECT DETECTION

To demonstrate the robustness of DW-MALA in a more complex task, we conducted experiments on the object detection task, which requires both the bounding box coordinates and class label of each object within an image. Accordingly, the training objective of the task model combines a bounding-box regression loss with a classification loss. We used PASCAL VOC 2007 and 2012 benchmarks (Everingham et al., 2010), consisting of 5,011 and 4,952 training images across 20 object categories, respectively. For the detection model, we adopted the Single Shot Multibox Detector (SSD) (Liu et al., 2016). To build the labeled dataset for training, we initially selected 1,000 images at random and added 1,000 newly acquired instances per iteration, resulting in 10,000 labeled instances at the final acquisition iteration. The model was trained for 300 epochs using the Adam (Kingma & Ba, 2014) optimizer with a learning rate of 1e-3 and a batch size of 32. Figure 7 reports

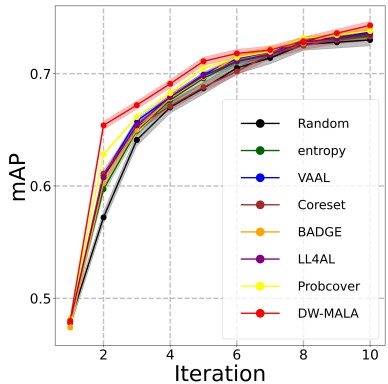

Figure 7: Object Detection Results

the mean average precision (mAP) of DW-MALA and the baselines along five independent trials. The final mAP scores are as follows: Random (73.0%), Coreset (73.4%), VAAL(73.7%), BADGE (73.9%), LL4AL (73.6%), ProbCover (73.8%) and DW-MALA (74.1%). Notably, DW-MALA achieves a significant improvement over baselines in early acquisition iterations.

### 4.4 DOMAIN ADAPTIVE SEMANTIC SEGMENTATION

Recently, active learning has been adopted for domain-adaptive semantic segmentation tasks (Xie et al., 2022). In our study, we addressed semantic segmentation that adapts models from a source domain of SYNTHIA dataset (Ros et al., 2016) to a target domain of CityScapes dataset (Cordts et al., 2016). As a baseline, we adopted the Region Impurity and Prediction Uncertainty (RI-PU) (Xie et al., 2022) method which queries regions centered around pixels with high scores that are calculated by multiplying diversity and uncertainty. Following the baseline, we propose RI-MALA, which computes region-based acquisition scores

| Method | mIOU |
|---|---|
| Random | 68.31 ±0.6 |
| RI-BADGE | 70.09 ±0.3 |
| RI-ProbCover | 70.23 ±0.2 |
| RI-PU | 70.32 ±0.4 |
| **RI-MALA** | **70.57 ±0.4** |

Table 3: Semantic Segmentation Results

as the product of diversity and representativeness calculated by DW-MALA. For active learning setting, 2.2% of pixels are used as labeled information for each image instance. Then, we select additional 2.2% of pixels, and the selection is repeated until a total of 11% of pixels are labeled. The model is trained using the Adam optimizer with a learning rate of 2.5e-4 and a batch size of 2, and we repeated the experiment with 5 independent trials. As shown in Table 3, RI-MALA achieves superior mIOU performance compared to baseline methods.

## 5 CONCLUSION

In this paper, we propose Diversity-Weighted Metropolis-Adjusted Langevin Algorithm, or DW-MALA, which is a novel hybrid active learning method combining representativeness and diversity. DW-MALA leverages the MALA algorithm to approximate the data distribution based on Langevin dynamics with log-gradient updates. Such dynamics of the MALA sampler enable us to approximate a precise data distribution so that we can find the major modes across the entire unlabeled data distribution. Based on the data distribution approximated by MALA, each unlabeled instance is assigned a representativeness score using its estimated density. In addition, DW-MALA also quantifies a diversity score via the distance to the nearest labeled instance. The resulting joint acquisition function of DW-MALA effectively balances the exploitation of high-density regions and the exploration of low-density regions. Extensive experiments across multiple tasks demonstrate that DW-MALA consistently outperforms baselines, particularly maximizing the gap in early acquisition iterations for an efficient use of annotation budget. Qualitative analyses further support the effectiveness of DW-MALA in selecting both representative and diverse instances. A potential limitation of DW-MALA is the computational cost of MALA sampling, which relies on gradient information that may be difficult to obtain in regions with non-differentiable structures. We leave addressing this challenge as an important direction for future work.

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

## A  DETAILED DERIVATION OF ANALYTIC SOLUTION FOR LANGEVIN SDE

For a particle of mass $m$, we first write the Langevin equation of Eq.(3) again as:

$$m\frac{d^2 x_t}{dt^2} = -\gamma\frac{dx_t}{dt} - \nabla U(x_t) + \sqrt{2}\frac{dW_t}{dt}, \tag{19}$$

where $x_t$ is the position of the particle at time $t$; $\gamma$ is the friction coefficient; $U(x_t)$ is the potential function whose gradient represents the deterministic force; and $W_t$ denotes a Wiener process (i.e., standard Brownian motion) whose gradient results in a stochastic force term, $\sqrt{2}\frac{dW_t}{dt}$.

Following the prior work of (Pavliotis, 2014), we assume that our dynamics follow the overdamped Langevin dynamics, which satisfy the condition of $\frac{m}{\gamma} \approx 0$. Under this condition, $\gamma \approx \infty$ becomes out of consideration because it indicates that the object is fixed without dynamics. Hence, we assume $m \approx 0$. Then, the Langevin equation is simplified as Eq.(4) as:

$$\frac{dx_t}{dt} = -\frac{1}{\gamma}\nabla U(x_t) + \frac{\sqrt{2}}{\gamma}\frac{dW_t}{dt}. \tag{20}$$

Since it is a complicated problem to solve, we assume that $U(x) = \frac{1}{2}kx^2$ for an arbitrary constant, $k$, and obtain Eq.(5) as:

$$\frac{dx_t}{dt} = -\frac{k}{\gamma}x_t + \frac{\sqrt{2}}{\gamma}\frac{dW_t}{dt}. \tag{21}$$

By introducing $u = -\frac{k}{\gamma}$ and $\sigma = \frac{\sqrt{2}}{\gamma}$, we get the simplified version as:

$$dx_t = ux_t dt + \sigma dB_t, \tag{22}$$

Then, we set the integrating factor as $e^{-ut}$ and get the derivations as:

$$e^{-ut}\, dx_t = e^{-ut}\big(ux_t\, dt + \sigma\, dB_t\big) \tag{23}$$

$$e^{-ut}\, dx_t - e^{-ut}ux_t dt = \sigma e^{-ut}\, dB_t \tag{24}$$

$$d\big(e^{-ut}x_t\big) = \sigma e^{-ut}\, dB_t \tag{25}$$

$$\int_0^t d\big(e^{-us}x_s\big) = \sigma \int_0^t e^{-us}\, dB_s \tag{26}$$

$$e^{-ut}x_t - x_0 = \sigma \int_0^t e^{-us}\, dB_s \tag{27}$$

Here, we represent the derivative of the Wiener process, $\frac{dW_t}{dt}$, as $dB_t$. Then, we get the final solution of Eq.(7) as:

$$x_t = x_0 e^{ut} + \sigma e^{ut}\int_0^t e^{-us}dB_s. \tag{28}$$

Under $U(x) = log(p(x))$ along with the Boltzmann distribution assumption $\pi_\beta \propto e^{-\beta U(x)}$; the Langevin dynamics can finally estimate a target distribution $p(x)$, which is the entire unlabeled data distribution in the active learning scenario.

# B  PSEUDO CODE OF DW-MALA

---

**Algorithm 1** DW-MALA: Diversity-Weighted MALA for Active Learning

---

**Input:**  Unlabeled dataset, $\mathcal{D}_U$; Labeled dataset, $\mathcal{D}_L$; Initial task model, $f$; Proposal distribution, $q(x)$; Number of acquisitions, $N$; Number of instances to select per iteration, $K$; Bandwidth for KDE, $h$; Number of samples proposed by MALA, $M$; Kernel function, $\Phi$; Score mixing coefficient, $\alpha$

**Output:** Optimized task model, $f$

1  **for** $t = 1$ *to* $N$ **do**

2      Initialize the sample set for MALA as $\mathcal{S}_{mala} = \emptyset$

3      **for** $i = 1$ *to* $M$ **do**

4          Propose new state $x_i$ by Eq.(11)

5          Calculate acceptance probability by Eq.(12)

6          Expand $\mathcal{S}_{mala}$ as $\mathcal{S}_{mala} \cup x_i$

7      **end**

8      Fit Kernel function $\Phi$ with bandwidth $h$ using $\mathcal{S}_{mala}$ as Eq.(15)

9      For each unlabeled data $x_u \in \mathcal{D}_U$, compute representativeness score $\mathcal{A}_{rep}(x_u)$:

$$\mathcal{A}_{rep}(x_u) = \frac{1}{Mh} \sum_{x_i \in \mathcal{S}_{mala}} \Phi\left(x_u - x_i\right)$$

10     For each unlabeled data $x_u \in \mathcal{D}_U$, compute diversity score $\mathcal{A}_{div}(x_u)$:

$$\mathcal{A}_{div}(x_u) = \min_{x_l \in \mathcal{D}_L} \text{Dist}(x_u, x_l)$$

11     Combine representativeness and diversity scores to compute joint score $\mathcal{A}_{DWMALA}(x_u)$:

$$\mathcal{A}_{DWMALA}(x_u) = \mathcal{A}_{div}(x_u)^{\alpha} \cdot \mathcal{A}_{rep}(x_u)^{1-\alpha}$$

12     Select top-$K$ instances from $\mathcal{D}_U$ based on $\mathcal{A}_{DWMALA}(x_u)$

13     Annotate the selected instances and move them from $\mathcal{D}_U$ to $\mathcal{D}_L$

14     Retrain the task model $f$ on the updated $\mathcal{D}_L$

15 **end**

---

# C ROBUSTNESS OF DW-MALA WITH VARIOUS VARIANTS

## C.1 CHOICE OF EUCLIDEAN DISTANCE

We mainly adopt Euclidean distance because it is the most intuitive metric and offers low computational overhead, which is especially important for repeated acquisition iterations in active learning. To further validate this choice, we conducted additional experiments on CIFAR-10 by replacing Euclidean distance with other metrics such as cosine similarity and Learned Perceptual Image Patch Similarity (LPIPS) (Zhang et al., 2018). Also, we extended our algorithm from single nearest neighbor to k-nearest neighbors by varying $k = 10, 20, 30$. The results in Table 4 show that the alternative distance metrics of cosine similarity and LPIPS yield no significant performance gain, while requiring longer computation time. Also, increasing $k$ in k-nearest neighbors similarly leads to minimal improvement but rather increased runtime cost. Given these findings, we conclude that using Euclidean distance to the single nearest labeled instance provides the best trade-off between efficiency and performance, which justifies our choice in the main experiments. We will include these additional results and clarifications in the final version of the paper.

| Distance Metric | 1 | 2 | 3 | 4 | 5 | 6 | 7 | 8 | 9 | 10 | sec |
|---|---|---|---|---|---|---|---|---|---|---|---|
| Cosine | 0.4486 | **0.6203** | 0.6410 | 0.7298 | 0.7316 | 0.7597 | 0.7716 | **0.7866** | 0.7910 | 0.8035 | 57162 |
| LPIPS | 0.4472 | 0.5568 | 0.6107 | 0.6922 | 0.7270 | 0.7462 | 0.7665 | 0.7739 | 0.7805 | 0.8026 | 62749 |
| K-nearest, k=10 | 0.4475 | 0.5580 | 0.5871 | 0.6788 | 0.7033 | 0.7168 | 0.7579 | 0.7490 | 0.7743 | 0.7915 | 57085 |
| K-nearest, k=20 | 0.4460 | 0.5389 | 0.6057 | 0.7000 | 0.7034 | 0.7551 | 0.7556 | 0.7885 | 0.7891 | 0.8013 | 59546 |
| K-nearest, k=30 | 0.4471 | 0.6136 | 0.6792 | 0.6964 | 0.7318 | 0.7479 | 0.7691 | 0.7682 | 0.7878 | 0.8024 | 61838 |
| Euclidean | 0.4490 | **0.6899** | **0.7225** | **0.7455** | **0.7591** | **0.7660** | **0.7751** | 0.7810 | **0.7951** | **0.8040** | 56923 |

Table 4: Test Accuracy on CIFAR-10 for Various Distance Metrics

## C.2 CHOICE OF MULTIPLICATIVE COMBINATION FOR HYBRID STRATEGY

The core idea of our approach is to select instances for which both representativeness and diversity scores are high, and Eq.(18) adopts a multiplicative combination inspired by prior hybrid active learning methods (Wang et al., 2021) that combine multiple acquisition criteria. However, we do not claim that this specific functional form is the only valid choice. To verify this, we performed additional experiments by replacing the combination function with linear combination, summation, and multiplication variants. We found that the results in Table 5 showed robustness without significant differences, while outperforming baselines regardless of the specific combination function. This indicates that the key factor is jointly considering both scores, rather than the exact combination form.

| Combination Function | 1 | 2 | 3 | 4 | 5 | 6 | 7 | 8 | 9 | 10 |
|---|---|---|---|---|---|---|---|---|---|---|
| $\mathcal{A}_{div}(x_u)^{\alpha} + \mathcal{A}_{rep}(x_u)^{1-\alpha}$ | 0.4468 | 0.6551 | 0.6817 | 0.7376 | 0.7454 | 0.7628 | 0.7734 | **0.7838** | 0.7930 | 0.8037 |
| $\mathcal{A}_{div}(x_u) + \mathcal{A}_{rep}(x_u)$ | 0.4466 | 0.6016 | 0.6401 | 0.7055 | 0.7298 | 0.7430 | 0.7665 | 0.7680 | 0.7833 | 0.7994 |
| $\mathcal{A}_{div}(x_u) \cdot \mathcal{A}_{rep}(x_u)$ | 0.4452 | 0.5512 | 0.6012 | 0.6903 | 0.7112 | 0.7393 | 0.7600 | 0.7705 | 0.7813 | 0.8005 |
| $\mathcal{A}_{div}(x_u)^{\alpha} \cdot \mathcal{A}_{rep}(x_u)^{1-\alpha}$ | 0.4490 | **0.6899** | **0.7225** | **0.7455** | **0.7591** | **0.7660** | 0.7751 | **0.7810** | **0.7951** | 0.8040 |

Table 5: Test Accuracy on CIFAR-10 for Various Combination Methods

# D  TRADE-OFF BETWEEN MALA SAMPLE SET SIZE AND PERFORMANCE

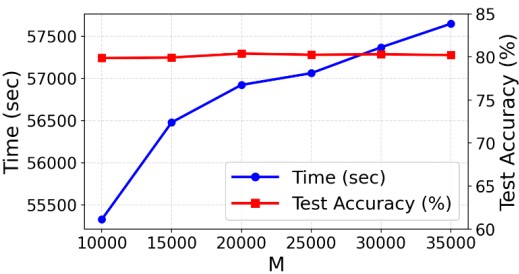

Figure 8: Trade-off Analysis of DW-MALA

To show the robustness of DW-MALA with regard to the MALA sample size, $M$, we further conducted experiments with DW-MALA by varying $M$ over $\{10000, 15000, 20000, 25000, 30000, 35000\}$. As shown in Figure 8, the time complexity grows with $M$, while maintaining test accuracy similarly across all the $M$ values. The default value in the main experiments was $M = 20000$, which shows the best test accuracy among the candidate $M$ values. However, it is worth noting that decreasing $M$ to 10000 would not affect its test accuracy but would rather result in a more efficient complexity.

# E  UMAP ANALYSIS OF DW-MALA AND BASELINES

Following Figures 5a − 5b in Section 4.2.2, we further provide the UMAP analysis on DW-MALA and the baselines of Entropy, BADGE, Coreset and DWDS; for the entire acquisition iterations except the initial iteration, where the same initial labeled dataset is shared. The following figures describe the selection process of all the methods, where we represent the labeled instances as colored dots, the unlabeled instances as gray dots, and the selected instances as black triangles.

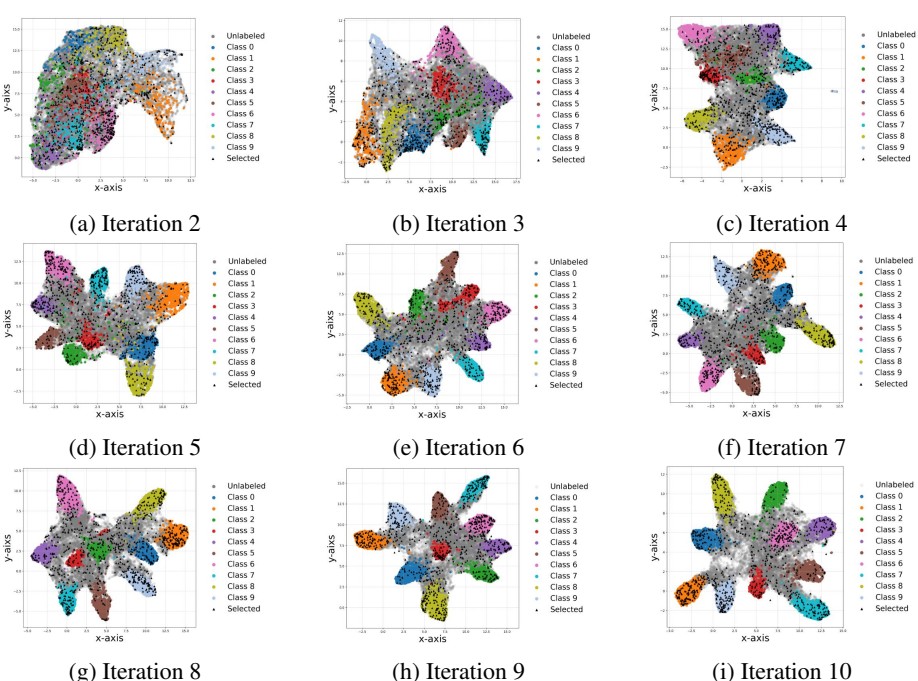

Figure 9: UMAP of Selection Process of *DW-MALA* on CIFAR-10

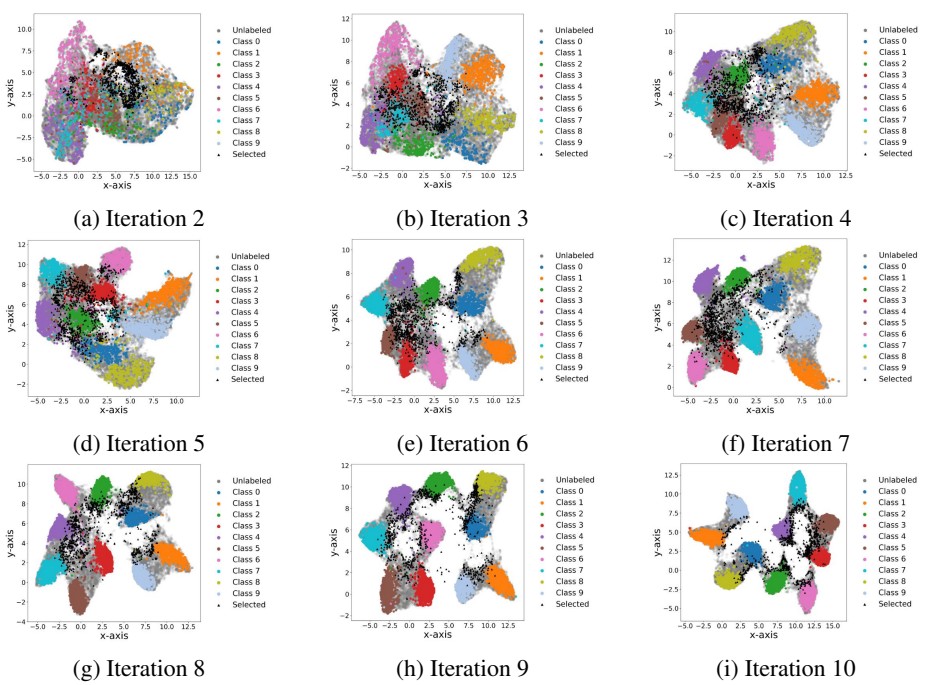

Figure 10: UMAP of Selection Process of *Entropy* on CIFAR-10

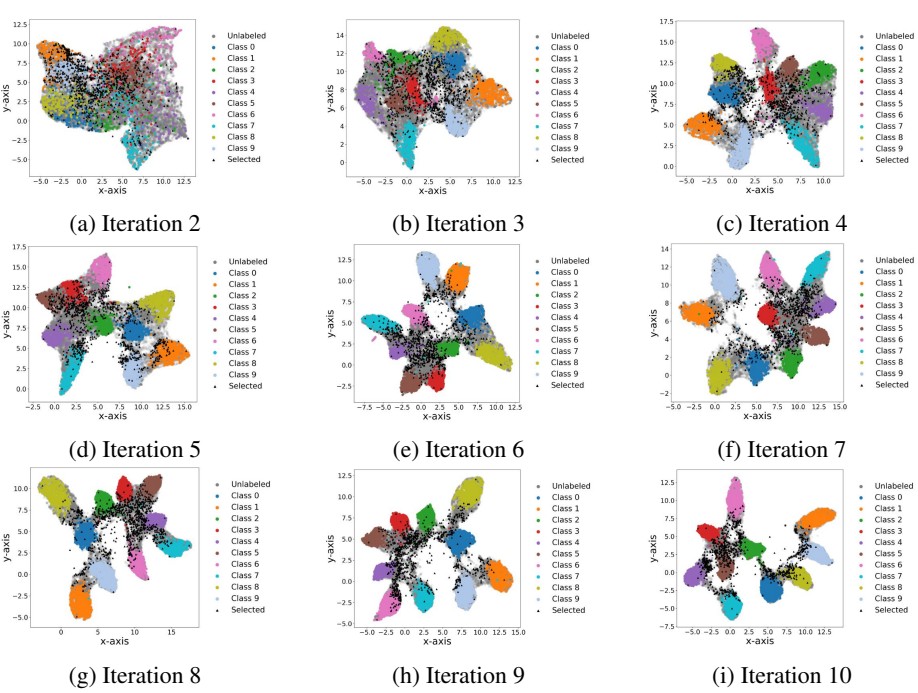

Figure 11: UMAP of Selection Process of *BADGE* on CIFAR-10

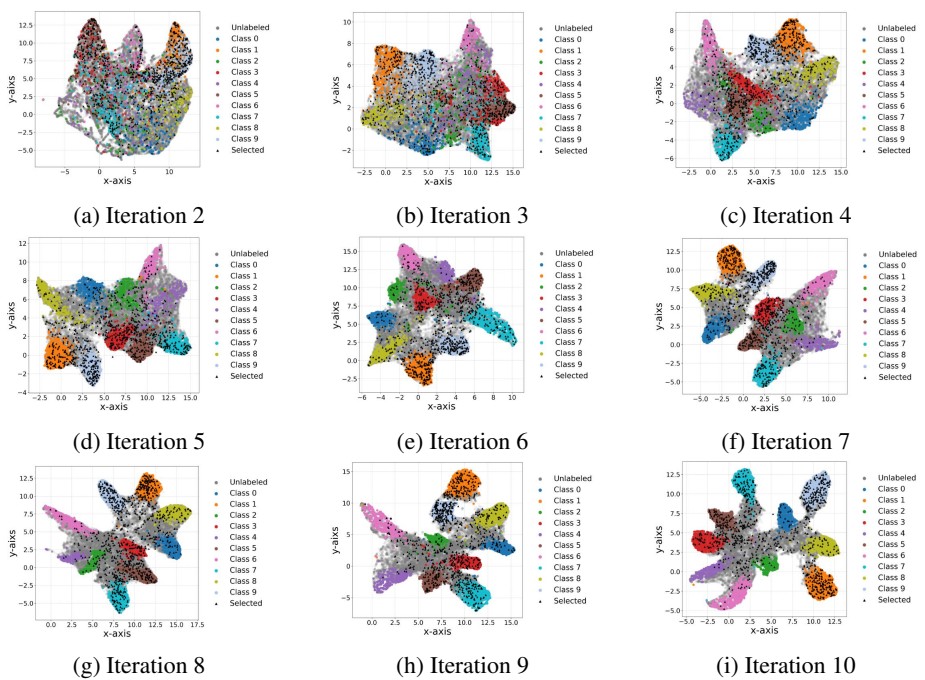

(a) Iteration 2      (b) Iteration 3      (c) Iteration 4

(d) Iteration 5      (e) Iteration 6      (f) Iteration 7

(g) Iteration 8      (h) Iteration 9      (i) Iteration 10

Figure 12: UMAP of Selection Process of *Coreset* on CIFAR-10

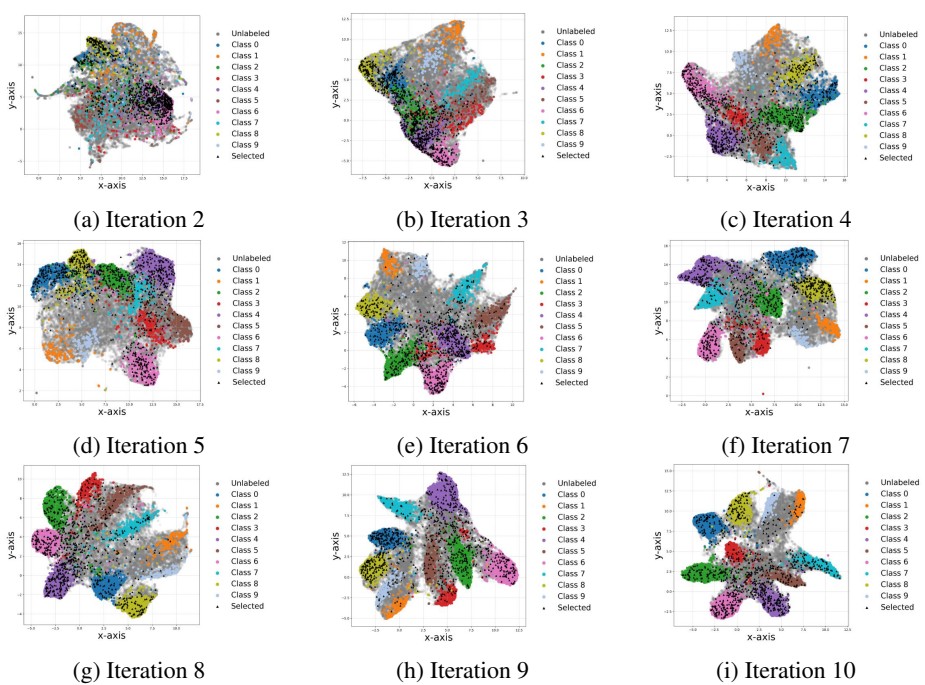

(a) Iteration 2      (b) Iteration 3      (c) Iteration 4

(d) Iteration 5      (e) Iteration 6      (f) Iteration 7

(g) Iteration 8      (h) Iteration 9      (i) Iteration 10

Figure 13: UMAP of Selection Process of *DWDS* on CIFAR-10

Entropy baseline tends to select instances from overlapping regions, since the instances on which the task model shows high entropy will have similar characteristics. BADGE tries to select instances in a diverse way, but some modes are neglected because there is no guarantee of diversity in the data space when using the gradient embedding as the input of k-Means++ seeding algorithm. DWDS and Coreset seem to select diverse instances, but DWDS uses cosine similarity as representativeness score and selects some instances that overlap in certain modes. In contrast, Coreset covers a broader region than DWDS, but some modes are still neglected from the search. Finally, our DW-MALA selects diverse and representative instances compared to the baselines.

## F  LABEL DISTRIBUTION OF SELECTED INSTANCES

Regarding Figure 5c in Section 4.2.2, we provide the detailed label distribution of selected instances at each acquisition iteration. That is, each numerical result in Figure 5c is calculated based on the distributions of Figure 14 below.

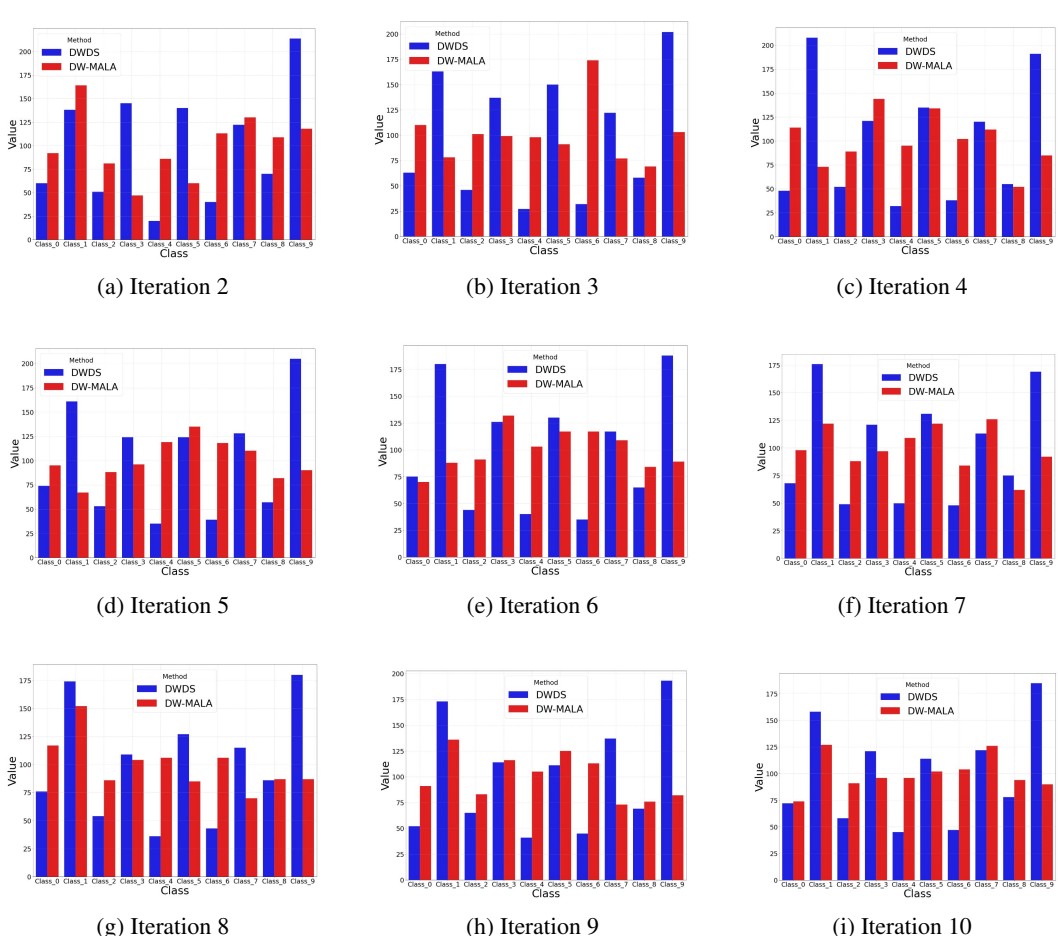

| (a) Iteration 2 | (b) Iteration 3 | (c) Iteration 4 |
| (d) Iteration 5 | (e) Iteration 6 | (f) Iteration 7 |
| (g) Iteration 8 | (h) Iteration 9 | (i) Iteration 10 |

Figure 14: Label Distribution of Selected Instances by DWDS and DW-MALA

As shown in the figures, DWDS selects instances whose labels are biased; for example, instances of class 1 or class 9 are commonly selected across all the iterations. However, in our DW-MALA, though the majority of selected labels change as the iteration proceeds, the gap between majority and minority is slighter compared to DWDS; and the selected instances construct a balanced label distribution in total.

## G CODE IMPLEMENTATION

We provide our implementation codes as 1) a zip file in the Supplementary material and 2) the anonymous GitHub link of `https://github.com/AnonymousForConference/ICLR2026_DWMALA`.

