# OpenReview forum: "Hybrid Query Strategy with Diversity-Weighted Metropolis–Adjusted Langevin Algorithm"
_ICLR.cc/2026/Conference — Submitted to ICLR 2026_

### Official Review · Reviewer_SnKu · 2025-10-31

**Soundness:** 3
**Presentation:** 3
**Contribution:** 2
**Rating:** 4
**Confidence:** 4

**Summary:**

This paper proposes a new AL approach that selects samples based on both diversity and data-distribution representativeness. Instead of relying only on uncertainty or distance-based heuristics, the approach uses Metropolis-Adjusted Langevin Algorithm (MALA) to approximate the underlying data density in the feature space, and identify points that better reflect the overall structure of the unlabeled data pool. Experiments on classification, imbalanced settings, object detection, and domain-shift segmentation tasks/data situations show competitive performance, especially in early AL rounds.

**Strengths:**

This paper presents a technically interesting AL method by incorporating Langevin-based density estimation into the AL sample selection process, this method aims to capture data representativeness. The idea of leveraging MALA in this context is interesting, it is a creative combination of sampling-based density modeling with hybrid acquisition functions.
The experimental evaluation is comprehensive across multiple datasets and task types, and the results support the method’s strengths, particularly in early AL rounds. The presentation is generally clear and easy to follow. In terms of significance, the method tries to address the limitation of existing diversity-based AL approaches and demonstrates practical gains.

**Weaknesses:**

I have some concerns:
1. The method is evaluated only under fixed random initialization; it does not study the effect of smaller or imbalanced seed sets, nor self-supervised warm-starts. However, the proposed method relies heavily on the quality of early learned features to perform MALA sampling. If initial embeddings are weak or even biased distribution, the estimates would drift and amplify bias across AL rounds.
2. The method considers the representativeness and diversity but doesn't include uncertainty, but uncertainty is crucial for boundary exploration in many AL situations. Although DWUS was tested, it even underperforms, and the paper does not analyze why or explore alternative integration ways.
3. This paper provided sensitivity studies show the method is not too sensitive to $\alpha$ or $M$. However, other hyperparameters like Langevin and KDE hyperparameters (e.g., step size, noise scale, kernel bandwidth) are not examined.

**Questions:**

1. The improvements on semantic segmentation tasks are marginal. Could the author explain more about why the proposed method does not perform competitively on such dense prediction tasks?

2. Figure 13 suggests the embedding space is weak early and improves gradually, yet DW-MALA yields the largest gains in early rounds. Could the authors elaborate on why density-driven sampling performs best specifically when feature quality is lowest?

---

> ### Author Response · Authors · 2025-11-21
> **Responses to Reviewer SnKu (1)**
>
> We feel grateful for the constructive reviews. We provide responses to the reviews as well as supporting experimental results below.
>
> 1. Imbalanced initial seed sets
>
> To evaluate the robustness of DW-MALA under more challenging initialization conditions, we re-conducted the classification experiment with an imbalanced initial labeled set, which follows the long-tail setting. This setting further degrades the quality of early feature embeddings, as the classifier becomes biased toward majority classes. The results in the table below confirm that DW-MALA continues to outperform the baselines with particularly large gains in the early acquisition iterations, indicating that DW-MALA is less sensitive to early-stage feature instability.
>
> - Experiments on CIFAR-10
>
> | Method | 1| 2 | 3| 4 | 5  | 6  | 7| 8| 9  | 10 |
> |-|-|-|-|-|-|-|-|-|-|-|
> | Random| $0.2542 \pm 0.0019$| $0.4217 \pm 0.0024$| $0.5344 \pm 0.0018$ | $0.6107 \pm 0.0021$| $0.6488 \pm 0.0012$ | $0.6821 \pm 0.0014$| $0.6942 \pm 0.0023$ | $0.7360 \pm 0.0013$| $0.7367 \pm 0.0017$ | $0.7468 \pm 0.0011$|
> | Entropy| $0.2551 \pm 0.0032$| $0.4234 \pm 0.0027$| $0.5297 \pm 0.0023$ | $0.6115 \pm 0.0028$| $0.6499 \pm 0.0032$ | $0.6846 \pm 0.0022$| $0.6965 \pm 0.0015$| $0.7333 \pm 0.0015$ | $0.7442 \pm 0.0023$| $0.7472 \pm 0.0029$ |
> | Coreset   | $0.2545 \pm 0.0027$| $0.4265 \pm 0.0011$| $0.5371 \pm 0.0021$| $0.6142 \pm 0.0015$| $0.6524 \pm 0.0032$ | $0.6870 \pm 0.0012$ | $0.7003 \pm 0.0032$| $0.7321 \pm 0.0012$ | $0.7455 \pm 0.0014$| $0.7475 \pm 0.0027$|
> | VAAL| $0.2547 \pm 0.0024$| $0.4391 \pm 0.0032$| $0.5387 \pm 0.0017$| $0.6146 \pm 0.0022$| $0.6509 \pm 0.0029$| $0.6857 \pm 0.0031$ | $0.6973 \pm 0.0028$| $0.7348 \pm 0.0018$| $0.7431 \pm 0.0029$ | $0.7477 \pm 0.0027$ |
> | LL4AL | $0.2551 \pm 0.0014$  | $0.4297 \pm 0.0029$| $0.5351 \pm 0.0024$| $0.6134 \pm 0.0024$| $0.6512 \pm 0.0018$ | $0.6858 \pm 0.0017$| $0.6980 \pm 0.0032$ | $0.7334 \pm 0.0020$ | $0.7443 \pm 0.0013$ | $0.7473 \pm 0.0028$|
> | BADGE| $0.2549 \pm 0.0014$ | $0.4501 \pm 0.0016$ | $0.5459 \pm 0.0014$ | $0.6205 \pm 0.0012$ | $0.6536 \pm 0.0013$ | $0.6904 \pm 0.0026$| $0.7011 \pm 0.0031$ | $0.7348 \pm 0.0017$ | $0.7482 \pm 0.0033$| $0.7488 \pm 0.0013$|
> | ProbCover | $0.2544 \pm 0.0012$| $0.4281 \pm 0.0015$| $0.5402 \pm 0.0017$| $0.6194 \pm 0.0024$| $0.6530 \pm 0.0026$| $0.6899 \pm 0.0018$ | $0.7008 \pm 0.0024$| $0.7345 \pm 0.0029$ | $0.7478 \pm 0.0028$ | $0.7483 \pm 0.0019$|
> | DW-MALA| $0.2549 \pm 0.0030$| $0.4721 \pm 0.0015$  | $0.5516 \pm 0.0019$ | $0.6217 \pm 0.0015$  | $0.6541 \pm 0.0021$ | $0.6904 \pm 0.0022$  | $0.7012 \pm 0.0031$ | $0.7351 \pm 0.0019$  | $0.7485 \pm 0.0015$| $0.7492 \pm 0.0014$|
>
> - Experiments on CIFAR-100
>
> | Method|1|2|3|4|5|6|
> |-|-|-|-|-|-|-|
> | Random| $0.1604 \pm 0.0034$| $0.2583 \pm 0.0025$| $0.3175 \pm 0.0035$| $0.3592 \pm 0.0025$ | $0.3786 \pm 0.0021$| $0.3811 \pm 0.0015$|
> | Entropy| $0.1608 \pm 0.0032$| $0.2597 \pm 0.0028$| $0.3174 \pm 0.0037$| $0.3605 \pm 0.0030$| $0.3791 \pm 0.0033$                    | $0.3820 \pm 0.0026$|
> | CoreSet   | $0.1610 \pm 0.0024$| $0.2589 \pm 0.0035$| $0.3167 \pm 0.0033$| $0.3614 \pm 0.0031$| $0.3799 \pm 0.0027$                    | $0.3826 \pm 0.0034$|
> | VAAL | $0.1610 \pm 0.0021$| $0.2673 \pm 0.0026$|$0.3195 \pm 0.0035$| $0.3625 \pm 0.0028$ | $0.3805 \pm 0.0015$                    | $0.3857 \pm 0.0020$ |
> | LL4AL| $0.1607 \pm 0.0026$| $0.2594 \pm 0.0021$| $0.3181 \pm 0.0019$| $0.3599 \pm 0.0019$ | $0.3788 \pm 0.0031$ | $0.3816 \pm 0.0022$|
> | BADGE| $0.1609 \pm 0.0024$| $0.2655 \pm 0.0032$| $0.3191 \pm 0.0031$| $0.3619 \pm 0.0036$| $0.3804 \pm 0.0024$| $0.3847 \pm 0.0034$|
> | ProbCover | $0.1615 \pm 0.0032$| $0.2612 \pm 0.0029$ | $0.3194 \pm 0.0035$|$0.3631 \pm 0.0025$|$0.3806 \pm 0.0033$| $0.3872 \pm 0.0019$|
> | DW-MALA   | $0.1606 \pm 0.0022$| $0.2809 \pm 0.0021$ | $0.3217 \pm 0.0030$|$0.3640 \pm 0.0034$| $0.3818 \pm 0.0030$| $0.3878 \pm 0.0027$|
>
> - Experiments on Tiny-ImageNet
>
> | Method|1|2|3|4|5
> |-|-|-|-|-|-|
> | Random| $0.0043 \pm 0.0020$ | $0.1136 \pm 0.0030$| $0.1621 \pm 0.0030$| $0.1875 \pm 0.0021$| $0.1946 \pm 0.0036$|
> | Entropy| $0.0042 \pm 0.0022$| $0.1148 \pm 0.0036$| $0.1633 \pm 0.0022$ | $0.1872 \pm 0.0033$| $0.1953 \pm 0.0031$                    |
> | CoreSet| $0.0043 \pm 0.0033$| $0.1146 \pm 0.0020$| $0.1635 \pm 0.0031$| $0.1879 \pm 0.0019$| $0.1953 \pm 0.0034$                    |
> | VAAL | $0.0043 \pm 0.0030$| $0.1153 \pm 0.0036$| $0.1647 \pm 0.0020$| $0.1888 \pm 0.0018$| $0.1960 \pm 0.0035$                    |
> | LL4AL| $0.0042 \pm 0.0024$| $0.1152 \pm 0.0030$| $0.1646 \pm 0.0017$| $0.1890 \pm 0.0017$| $0.1959 \pm 0.0027$|
> | BADGE| $0.0041 \pm 0.0022$|$0.1217 \pm 0.0017$| $0.1653 \pm 0.0019$| $0.1890 \pm 0.0028$| $0.1962 \pm 0.0036$|
> | ProbCover | $0.0040 \pm 0.0028$| $0.1168 \pm 0.0025$|$0.1669 \pm 0.0022$|$0.1905 \pm 0.0034$ |$0.1972 \pm 0.0026$|
> | DW-MALA   | $0.0039 \pm 0.0019$| $0.1412 \pm 0.0035$ | $0.1697 \pm 0.0025$| $0.1910 \pm 0.0028$| $0.1978 \pm 0.0021$|

---

> ### Author Response · Authors · 2025-11-21
> **Responses to Reviewer SnKu (2)**
>
> 2. Considering uncertainty
>
> We appreciate this valuable observation. We agree that uncertainty plays an important role in boundary exploration for active learning. The relatively low performance of DWUS is likely due to its heuristic representativeness estimation, which relies on cosine similarity rather than an explicit density model. As a result, when feature embeddings are still weak, its representativeness score becomes unstable. Notably, since DW-MALA is fundamentally a hybrid acquisition framework, uncertainty can be readily incorporated. To examine this, we extended the acquisition score of DW-MALA in Eq. (18) by integrating entropy $H(x)$ as an additional multiplicative factor. The new acquisition score becomes: $A_{hybrid}(x)=A_{div}(x)^{\alpha}*A_{rep}(x)^{1−\alpha}*H(x)$. Experimental results in the table below demonstrate that this uncertainty-augmented DW-MALA still performs well across the entire acquisition process. This indicates that estimating a high-quality metric of each component is crucial for constructing a hybrid acquisition function.
> | Method          | 1       | 2       | 3          | 4       | 5       | 6       | 7       | 8       | 9       | 10      |
> |-----------------|---------|---------|------------|---------|---------|---------|---------|---------|---------|---------|
> |DW-MALA*Entropy $(A_{div}(x)^{\alpha}*A_{rep}(x)^{1−\alpha}*H(x))$ | 0.4488  | 0.6884  | 0.7230 | 0.7439  | 0.7559  | 0.7693  | 0.7740  | 0.7805  | 0.7906  | 0.8016  |
> | DW-MALA $(A_{div}(x)^{\alpha}*A_{rep}(x)^{1−\alpha})$         | 0.4490  | 0.6899 | 0.7225  | 0.7455 | 0.7591 | 0.7660 | 0.7751 | 0.7810 | 0.7951 | 0.8040  |
>
> While one could design more sophisticated combination strategies, the findings in Appendix C.2 already indicate that the key contributor is the accurate scoring of each criterion, rather than the specific mathematical form of their combination. Hence, we expect DW-MALA to remain robust under various integration formats. We will include this additional experiment and discussion to clarify that DW-MALA can naturally incorporate uncertainty and still demonstrate the best performance.
>
> 3. Sensitivity analysis on hyperparameters of Langevin and KDE
>
> According to the reviewer’s comment, we additionally conducted sensitivity analysis with step size $(\epsilon)$ in Eq.(11) and kernel bandwidth $(h)$ in Eq.(15). The candidate values for both hyperparameters were $\\{0.5, 1, 1.5, 2, 2.5\\}$. The table below shows the performance of DW-MALA across all the combinations of the hyperparameters, which reports mere differences among them.
>
> | $h$ / $\epsilon$ | 0.5    | 1      | 1.5    | 2      | 2.5    |
> |--------|--------|--------|--------|--------|--------|
> | 0.5    | 0.8030 | 0.8027 | 0.8024 | 0.8025 | 0.8021 |
> | 1      | 0.8040 | 0.8036 | 0.8031 | 0.8033 | 0.8038 |
> | 1.5    | 0.8031 | 0.8036 | 0.8033 | 0.8024 | 0.8028 |
> | 2      | 0.8035 | 0.8032 | 0.8031 | 0.8030 | 0.8027 |
> | 2.5    | 0.8023 | 0.8019 | 0.8026 | 0.8024 | 0.8029 |
>
> Regarding the MALA step size $\epsilon$, prior theoretical results ensure strong convergence guarantees for a wide range of step sizes, meaning performance variations tend to be limited. Empirically, we observed stable behavior within standard ranges.
> For the kernel bandwidth $h$, its influence is typically much weaker than the effect of the MALA sample size $M$, since $h$ only smooths density estimates on top of the already-distributed MALA proposals, as in Eq. (15). Having said that, since DW-MALA remains robust across a wide range of $M$ values (see Appendix D), the sensitivity of $h$ would be even smaller, and we confirmed this conjecture in the table.

---

> ### Author Response · Authors · 2025-11-21
> **Responses to Reviewer SnKu (3)**
>
> 4. Marginal improvement on semantic segmentation
>
> We agree that reporting only the final iteration results may under-emphasize the benefit of active learning, since the labeled dataset has already grown sufficiently large by the last iterations, making performance differences naturally converge. To address this concern, we will include the performance curves across all acquisition iterations in the revised manuscript, while we first provide the results in the table below. The newly added results show a performance trend that is consistent with the classification and detection experimental results, in that DW-MALA yields a clear advantage in the early acquisition rounds, where annotation efficiency and sample quality matter the most.
> | Method        | Iter10000           | Iter12000           | Iter14000           | Iter16000           | Iter18000           | Iter40000           |
> |--------------|---------------------|---------------------|---------------------|---------------------|---------------------|---------------------|
> | Random       | $56.46 \pm 0.4$     | $59.57 \pm 0.3$     | $61.07 \pm 0.5$     | $63.29 \pm 0.4$     | $66.61 \pm 0.5$     | $68.31 \pm 0.6$     |
> | RI-BADGE     | $58.40 \pm 0.2$     | $62.11 \pm 0.4$     | $64.20 \pm 0.3$     | $65.79 \pm 0.4$     | $68.88 \pm 0.3$     | $70.09 \pm 0.3$     |
> | RI-ProbCover | $58.65 \pm 0.4$     | $61.84 \pm 0.4$     | $64.25 \pm 0.3$     | $66.04 \pm 0.2$     | $68.96 \pm 0.5$     | $70.23 \pm 0.2$     |
> | RI-PU        | $58.59 \pm 0.2$     | $62.51 \pm 0.3$     | $64.48 \pm 0.3$     | $66.76 \pm 0.4$     | $69.31 \pm 0.3$     | $70.32 \pm 0.4$     |
> | RI-MALA      | **$61.08 \pm 0.4$** | **$63.05 \pm 0.1$** | **$64.72 \pm 0.2$** | **$66.87 \pm 0.3$** | **$69.42 \pm 0.3$** | **$70.57 \pm 0.4$** |
>
> 5. Effect of feature quality
>
> We appreciate the reviewer for raising this insightful question. We would like to clarify the underlying reason for this observation. Uncertainty-based or gradient-based acquisition strategies fundamentally rely on fine-grained feature separability or well-calibrated classifier confidence. However, in the early AL rounds, feature extractors are still immature, often producing noisy and unstable representations. This makes those approaches more susceptible to feature quality degradation. In contrast, DW-MALA is a density-driven method; that is, MALA sampling captures the coarse global structure of the feature manifold. The KDE-based representativeness metric focuses on cluster-level density, not boundary details, while the diversity metric ensures coverage of underexplored modes, independent of prediction confidence. Therefore, even when embeddings are still developing, DW-MALA remains robust by leveraging global density characteristics rather than fine-level discriminability, leading to larger gains in early acquisition rounds, as observed in various experiments. We will make this explanation clearer in the revised manuscript.

---

### Official Review · Reviewer_EKwH · 2025-10-31

**Soundness:** 3
**Presentation:** 2
**Contribution:** 3
**Rating:** 4
**Confidence:** 3

**Summary:**

This paper introduces DWMALA, an active learning strategy that combines two principles known in the active learning community which are representativeness and diversity when selecting samples for annotation. The proposal achieves representativeness by using the Metropolis-Adjusted Langevin Algorithm to sample candidate samples then weights the candidate samples using Kernel Density Estimation. The diversity is accounted for by using the minimum Euclidean distance between the candidate sample and the already labeled samples. The experiments on balanced and imbalanced datasets along with challenging tasks like object detection and semantic segmentation show improvements over other active learning strategies.

**Strengths:**

(S1) The integration of representativeness and diversity is clear, and the trade-off between the two components is systematically analyzed.
(S2) The method is evaluated across diverse settings and tasks, including classification, imbalanced data, object detection, and semantic segmentation, demonstrating broad applicability.
(S3) The paper includes thorough ablation studies and sensitivity analyses, providing good insight into the effect of design choices, particularly the α hyperparameter.

**Weaknesses:**

(W1) The use of a physical model used to model a dataset distribution is not motivated well.
(W2) The choice of the energy potential U(x) is not justified well. Why $$U(x) =\frac{1}{2} k x^2$$ ?
(W3) It is not mentioned how many runs are used for each experiment.
(W4) The improvements provided by the method are in many cases marginal.
(W5) The writing of the Ablation study section is a bit confusing. What is being held and what is used?

**Questions:**

(Q1) Please state how you will describe the motivation behind using a physical model to model the distribution of the datasets in the method section.
(Q2) Please explain how you will show statistically significance results in a revision of the manuscript. (Maybe it works well with imbalanced datasets but not good enough with balanced dataset)
(Q3) Please state how you would rewrite the ablation study section.

(Please specify how you will improve the paper as we reviewers are not interested in just private education but in improvements of the manuscript).

---

> ### Author Response · Authors · 2025-11-21
> **Responses to Reviewer EKwH (1)**
>
> We feel grateful for the constructive reviews. We provide responses to the reviews as well as supporting experimental results below.
>
> 1. Motivation of using physical model
>
> Our intention in adopting the physical model of Langevin-based formulation was to move beyond the heuristic representativeness estimation widely used in prior work, and instead explicitly model the underlying data distribution. The physical interpretation serves as a principled mathematical foundation that provides a well-established stochastic process whose stationary distribution matches the target density. This allows us to logically explain the sampling behavior by generating distribution-aware proposals. We also note that leveraging physical dynamics to model data distributions is gaining strong attention in modern AI research (e.g., diffusion models [1], physics-inspired PDEs [2]). Our work follows this emerging direction by adopting a probability-driven, physics-motivated approach for active learning.
>
> [1] Score-based Generative Modeling through Stochastic Differential Equations,( Y Song, et.al., Score-based generative modeling through stochastic differential equations, ICLR, 2021)
>
> [2] Poisson reweighted Lplacian Uncertainty Sampling for Graph-based Active Learning(K Miller, et.al., SIAM Journal on Mathematics of Data Science,2023)
>
> 2. Choice of energy potential
>
> The form $U(x)=1/2kx^2$ is introduced only up to Eq. (8) to illustrate the analytic derivation of the Langevin solution in a pedagogical manner. After transitioning to the discretized form, any arbitrary potential $U(x)$ can be applied. Thus, no quadratic assumption is used in the DW-MALA acquisition procedure.
>
> 3. Number of experiments
>
> We conducted 5 independent trials for all experiments, including classification, object detection, and domain adaptive semantic segmentation. This will be added explicitly in the revised manuscript.

---

> ### Author Response · Authors · 2025-11-21
> **Responses to Reviewer EKwH (2)**
>
> 4. Marginal improvements of DW-MALA
>
> We understand the reviewer’s concern that the overall numerical gains may appear moderate. However, we would like to highlight two important aspects of the results. First, DW-MALA demonstrates the largest accuracy increase in the earliest acquisition iterations, showing that our hybrid strategy uses the labeling budget more effectively than baselines. It should be noted that active learning emphasizes label efficiency, which is measured as the performance gains when annotations are still very limited, as in the early iterations. In later iterations, once ample labeled data is accumulated, convergence of performance is naturally expected across all methods. Second, DW-MALA maintains a steady performance advantage over the baselines, indicating robustness rather than relying on occasional favorable iterations. To quantify this robustness, we compared 1) winning frequency (i.e., the number of iterations where DW-MALA outperforms baseline over every experimental trial) and 2) the average performance margin (i.e., accuracy difference) across all possible pairs of methods. The results in the tables below show that DW-MALA outperforms every baseline in most acquisition iterations and also shows the largest improvement over the baseline.
>
> - Winning frequency at 2nd iteration
>
> |          | Random | Entropy | Coreset | VAAL | LL4AL | BADGE | ProbCover | DW-MALA |
> |----------|--------|---------|---------|------|-------|-------|-----------|---------|
> | Random   | 0.00   | 0.52    | 0.48    | 0.52 | 0.40  | 0.36  | 0.48      | 0.00    |
> | Entropy  | 0.48   | 0.00    | 0.20    | 0.52 | 0.36  | 0.40  | 0.36      | 0.00    |
> | Coreset  | 0.52   | 0.80    | 0.00    | 0.60 | 0.24  | 0.48  | 0.40      | 0.00    |
> | VAAL     | 0.48   | 0.48    | 0.40    | 0.00 | 0.36  | 0.40  | 0.40      | 0.00    |
> | LL4AL    | 0.60   | 0.64    | 0.76    | 0.64 | 0.00  | 0.56  | 0.56      | 0.00    |
> | BADGE    | 0.64   | 0.60    | 0.52    | 0.60 | 0.44  | 0.00  | 0.68      | 0.00    |
> | ProbCover| 0.52   | 0.64    | 0.60    | 0.60 | 0.44  | 0.72  | 0.00      | 0.00    |
> | DW-MALA  | 1.00   | 1.00    | 1.00    | 1.00 | 1.00  | 1.00  | 1.00      | 0.00    |
>
> - Performance (\%) margin at 2nd iteration
>
> |          | Random | Entropy | Coreset | VAAL | LL4AL | BADGE | ProbCover | DW-MALA |
> |----------|--------|---------|---------|------|-------|-------|-----------|---------|
> | Random   |  0.0   | -0.6    |  0.2    | -0.1 |  2.4  |  0.9  |  0.3      | -3.2    |
> | Entropy  |  0.6   |  0.0    |  0.8    |  0.5 |  3.1  |  1.5  |  0.9      | -2.5    |
> | Coreset  | -0.2   | -0.8    |  0.0    | -0.3 |  2.3  |  0.7  |  0.2      | -3.3    |
> | VAAL     |  0.1   | -0.5    |  0.3    |  0.0 |  2.6  |  1.0  |  0.5      | -3.0    |
> | LL4AL    | -2.4   | -3.1    | -2.3    | -2.6 |  0.0  | -1.5  | -2.1      | -5.6    |
> | BADGE    | -0.9   | -1.5    | -0.7    | -1.0 |  1.5  |  0.0  | -0.6      | -4.1    |
> | ProbCover| -0.3   | -0.9    | -0.2    | -0.5 |  2.1  |  0.6  |  0.0      | -3.5    |
> | DW-MALA  |  3.2   |  2.5    |  3.3    |  3.0 |  5.6  |  4.1  |  3.5      |  0.0    |
>
> - Winning frequency at final iteration
>
> |          | Random | Entropy | Coreset | VAAL | LL4AL | BADGE | ProbCover | DW-MALA |
> |----------|--------|---------|---------|------|-------|-------|-----------|---------|
> | Random   | 0.00   | 0.40    | 0.48    | 0.36 | 0.40  | 0.44  | 0.40      | 0.16    |
> | Entropy  | 0.60   | 0.00    | 0.40    | 0.40 | 0.32  | 0.40  | 0.28      | 0.08    |
> | Coreset  | 0.52   | 0.60    | 0.00    | 0.32 | 0.24  | 0.44  | 0.28      | 0.04    |
> | VAAL     | 0.60   | 0.60    | 0.68    | 0.00 | 0.36  | 0.40  | 0.40      | 0.08    |
> | LL4AL    | 0.60   | 0.68    | 0.72    | 0.64 | 0.00  | 0.56  | 0.52      | 0.12    |
> | BADGE    | 0.52   | 0.60    | 0.52    | 0.44 | 0.44  | 0.00  | 0.40      | 0.12    |
> | ProbCover| 0.60   | 0.72    | 0.68    | 0.60 | 0.44  | 0.60  | 0.00      | 0.08    |
> | DW-MALA  | 0.84   | 0.92    | 0.92    | 0.92 | 0.88  | 0.88  | 0.92      | 0.00    |
>
> - Performance (\%) margin at final iteration
>
> |          | Random | Entropy | Coreset | VAAL | LL4AL | BADGE | ProbCover | DW-MALA |
> |----------|--------|---------|---------|------|-------|-------|-----------|---------|
> | Random   |  0.0   | -0.3    | -0.3    | -0.6 | -0.6  | -0.4  | -0.7      | -3.9    |
> | Entropy  |  0.3   |  0.0    | -0.1    | -0.3 | -0.3  | -0.1  | -0.4      | -3.6    |
> | Coreset  |  0.3   |  0.1    |  0.0    | -0.3 | -0.3  | -0.1  | -0.5      | -3.7    |
> | VAAL     |  0.6   |  0.3    |  0.3    |  0.0 |  0.0  |  0.2  | -0.2      | -3.4    |
> | LL4AL    |  0.6   |  0.3    |  0.3    |  0.0 |  0.0  |  0.2  | -0.2      | -3.4    |
> | BADGE    |  0.4   |  0.1    |  0.1    | -0.2 | -0.2  |  0.0  | -0.3      | -3.5    |
> | ProbCover|  0.7   |  0.4    |  0.5    |  0.2 |  0.2  |  0.3  |  0.0      | -3.2    |
> | DW-MALA  |  3.9   |  3.6    |  3.7    |  3.4 |  3.4  |  3.5  |  3.2      |  0.0    |

---

> ### Author Response · Authors · 2025-11-21
> **Responses to Reviewer EKwH (3)**
>
> 5. Reorganizing ablation study
>
> Thank you for the constructive feedback. We agree that the current explanation of the ablation variants may be confusing, and we apologize for the lack of clarity. We will significantly improve the ablation study section to better communicate the experimental design and insights. The purpose of our ablation study is 1) to isolate the contribution of each acquisition component, which are diversity and representativeness; and 2) to compare heuristic representativeness estimation versus our explicit density-based representativeness. Having said that, we will add new ablation models, named DS (Diversity Sampling) and RS (Representativeness Sampling), which are clarified as the below:
> | Ablation Variants                 | Representativeness    | Diversty    |
> |-------------------------|-------|-----------|
> | DS       | X | O |
> | RS      | O (heuristic) | X |
> | DWDS     | O (heuristic) | O |
> | DW-MALA        | O (KDE-based) | O |
>
> Having said that, the experimental results at each iteration are as the table below.
> | Method  | 1                      | 2                          | 3                          | 4                          | 5                          | 6                          | 7                          | 8                          | 9                          | 10                         |
> |---------|------------------------|----------------------------|----------------------------|----------------------------|----------------------------|----------------------------|----------------------------|----------------------------|----------------------------|----------------------------|
> | DS  | $0.4452 \pm 0.0016$    | $0.6518 \pm 0.0027$        | $0.6706 \pm 0.0031$        | $0.7007 \pm 0.0033$        | $0.7088 \pm 0.0024$        | $0.7182 \pm 0.0028$        | $0.7349 \pm 0.0031$        | $0.7502 \pm 0.0027$        | $0.7582 \pm 0.0032$        | $0.7701 \pm 0.0025$        |
> | RS  | $0.4492 \pm 0.0021$    | $0.6505 \pm 0.0031$        | $0.6695 \pm 0.0016$        | $0.7002 \pm 0.0032$        | $0.6994 \pm 0.0022$        | $0.7164 \pm 0.0023$        | $0.7336 \pm 0.0015$        | $0.7452 \pm 0.0029$        | $0.7559 \pm 0.0033$        | $0.7679 \pm 0.0018$        |
> | DWDS    | $0.4458 \pm 0.0015$    | $0.6521 \pm 0.0036$        | $0.6911 \pm 0.0031$        | $0.7198 \pm 0.0030$        | $0.7345 \pm 0.0030$        | $0.7228 \pm 0.0034$        | $0.7464 \pm 0.0034$        | $0.7681 \pm 0.0020$        | $0.7559 \pm 0.0031$        | $0.7847 \pm 0.0033$        |
> | DW-MALA | $0.4490 \pm 0.0014$    | **$0.6899 \pm 0.0016$**    | **$0.7225 \pm 0.0009$**    | **$0.7455 \pm 0.0021$**    | **$0.7591 \pm 0.0019$**    | **$0.7660 \pm 0.0013$**    | **$0.7751 \pm 0.0018$**    | **$0.7810 \pm 0.0010$**    | **$0.7951 \pm 0.0019$**    | **$0.8041 \pm 0.0026$**    |
>
> Thus, the ablation study shows that 1) considering both representativeness and diversity score as a hybrid model benefits, by comparing DS and RS with DWDS and DW-MALA; and 2) density-based representativeness shows strong improvements over heuristic representativeness, by comparing DWDS and DW-MALA.
>
> 6. Statistical significance
>
> Following the reviewer’s guidance, we conducted t-tests across all acquisition iterations and against all baseline methods to assess the statistical reliability of performance gains. The null hypothesis is that both DW-MALA and the baseline show the same performance;, while the alternative hypothesis is that DW-MALA outperforms the baseline. The results are reported in the table below, which statistically proves that DW-MALA shows robust performance over baselines.
>
> | Comparing Method                 | T     | P-value    |
> |-------------------------|-------|-----------|
> | DW-MALA vs Random       | 4.138 | <0.001*** |
> | DW-MALA vs Entropy      | 4.191 | <0.001*** |
> | DW -MALA vs Coreset     | 3.776 | <0.001*** |
> | DW -MALA vs VAAL        | 3.471 | <0.001*** |
> | DW-MALA vs LL4AL        | 2.755 | 0.005**   |
> | DW-MALA vs BADGE        | 3.091 | <0.001*** |
> | DW-MALA vs ProbCover    | 2.900 | 0.003**   |

---

### Official Review · Reviewer_D34u · 2025-11-02

**Soundness:** 2
**Presentation:** 2
**Contribution:** 2
**Rating:** 2
**Confidence:** 4

**Summary:**

The paper proposes an active learning method based on Metropolis adjusted Langevin dynamics. Based on the estimated data distribution and its gradient flow, it draws reasonable samples from which it measures representativeness of candidate samples of active learning. In addition, the authors also adds diversity weighting based on distance in feature space of a classifier. They compare the proposed method on several active learning methods on some popular benchmark datasets for image classification as well as object detection and semantic segmentation tasks.

**Strengths:**

1. The paper proposes a novel active learning method based on actual density estimation of data distribution.
2. It is fairly easy to read the paper although some details are missing.
3. The proposed method was demonstrated on several tasks for image recognition such as classification, object detection and semantic segmentation.

**Weaknesses:**

1. The derivation of Langevin dynamics in Eq.(11) is based on some simplifications such as $U(x) = \frac{1}{2}kx^2$, $\beta=1$ and $\gamma=1$.
2. It seems like the target distribution $\pi$ is estimated as KDE with a Gaussian kernel. I do not think this is assumption is practically useful especially in high-dimensional space.
3. The paper says “most of the previous methods have focused on designing heuristic metrics” but I am not clear if the proposed DW-MALA is based on heuristics especially in terms of weighting using $\alpha$. Is there any theoretical justification of why the weighting has to be that way?
4. It is missing many highly relevant literature.
* Representation-based
    * Guy Hacohen, Avihu Dekel, and Daphna Weinshall. “Active learning on a budget: opposite strategies suit high and low budgets”, ICML 2022.
    * Wonho Bae, Junhyug Noh, and Danica J Sutherland. “Generalized coverage for more robust low-budget active learning”, ECCV 2024.
* Hybrid methods
    * Amin Parvaneh, Ehsan Abbasnejad, Damien Teney, Gholamreza Reza Haffari, Anton Van Den Hengel, and Javen Qinfeng Shi. “Active learning by feature mixing”, CVPR 2022.
    * Yichen Xie, Han Lu, Junchi Yan, Xiaokang Yang, Masayoshi Tomizuka, and Wei Zhan. “Active finetuning: exploiting annotation budget in the pretraining-finetuning paradigm”, CVPR 2023.
    * Guy Hacohen and Daphna Weinshall. “How to select which active learning strategy is best suited for your specific problem and budget”, NeurIPS 2024.
    * Wonho Bae, Gabriel L. Oliveira, Danica J. Sutherland, “Uncertainty herding: one active learning method for all label budgets”, ICLR 2025.
5. The experiments were conducted only on small size datasets: CIFAR10, 100 and Tiny Imagenet. It is unsure how the proposed method scales up to larger datasets like ImageNet.
6. Sensitivity analysis on $\alpha$ in Section 4.2.3 shows the differences across settings are marginal. It sounds like it does not really matter to just use either representativeness or diversity measure. Given that diversity measure is nothing new, it is not clear if the proposed method adds any significant improvement over the diversity baseline.

**Questions:**

1. According to the code shared by the authors, this is how Gaussian mixture samples are drawn. But, I do not understand how this “GMM” has any meaning without fitting GMM parameters. Aren’t they simply prior of GMM not posterior, which means there is no information from data?

```
def sample_from_gmm(num_samples, latent_dim='', num_modes=''):
    means = np.random.randn(num_modes, latent_dim) * 2.0
    cov = np.array([np.eye(latent_dim) for _ in range(num_modes)])
    samples = []
    for _ in range(num_samples):
        k = np.random.randint(0, num_modes)
        sample = np.random.multivariate_normal(means[k], cov[k])
        samples.append(sample)
    return np.array(samples)
```

2. How was wall-clock time measured for time complexity in Section 4.1.4? I do not particularly understand the fact that Random took 12.17sec. Even for Entropy 45.64 seems pretty high when the classifier is VGG.

---

> ### Author Response · Authors · 2025-11-21
> **Response to Reviewer D34u (1)**
>
> We feel grateful for the constructive reviews. We provide responses to the reviews as well as supporting experimental results below.
>
> 1. Simplification on derivations
>
> These simplifications follow standard MALA derivations [1,2]. Importantly, the form $U(x)=1/2kx^2$ is introduced only up to Eq.(8) to illustrate the analytic derivation of the Langevin solution in a pedagogical manner. After transitioning to the discretized form, any arbitrary potential $U(x)$ can be applied. Thus, no quadratic assumption is used in the DW-MALA acquisition procedure. Regarding $\beta$ and $\gamma$, we follow standard practice in prior MALA studies by setting both $\beta$=1 and $\gamma$=1 for notational simplicity. Since they are constant scaling factors, assigning different values would not materially affect the sampling behavior. We will revise Section 3.1-3.2 to explicitly state that these simplifications are for derivation clarity and do not restrict the generality of DW-MALA.
>
> [1] Statical Finite Elements via Langevin Dynamics (ÖD Akyildiz et al., SIAM/ASA Journal on Uncertainty Quantification, 2022)
>
> [2] Generative Modeling by Estimating Gradients of the Data Distribution (Y Song et.al., Neurips,2019)
>
> 2. Dimensionality of KDE
>
> We apologize for not clearly specifying this aspect in the manuscript. Our KDE is not computed in the raw pixel space, but rather in the penultimate-layer feature embedding space extracted from the task model. The embedding has significantly lower dimensionality than the original input image, which greatly reduces the risk of high-dimensional density estimation failure. We will incorporate these details into Section 3.3 and the experimental settings.
>
> 3. Meaning of heuristic metrics
>
> We apologize for the ambiguity in the original writing. Our comment about heuristic design was not referring to the weighting parameter α. Instead, we specifically addressed the representativeness estimation process itself, which was used in prior hybrid AL methods. For example, several baselines rely on heuristic clustering structures such as k-means or cosine similarity to approximate density, which can fail to reflect the underlying data distribution accurately. In contrast, DW-MALA explicitly models the target data density using MALA, then directly selects samples from high-density regions inferred from this estimation. Thus, our representativeness metric is not heuristic, but rather a principled and data-driven estimation of probability density on the feature manifold. Regarding alpha, our weight simply balances exploration and exploitation, while we demonstrated that the performance is robust to $\alpha$ variations (see Figure6) and the method remains strong under multiple joint-score formulation (see Appendix C.2). These results confirm that DW-MALA is not sensitive to hyperparameter tuning nor to a specific function combination.

---

> ### Author Response · Authors · 2025-11-21
> **Response to Reviewer D34u (2)**
>
> 4. Missing relevant literature
>
> Thank you for listing these works. All citations will be added to the revised manuscript, including discussions which compare with DW-MALA. Notably, DW-MALA uses explicit density estimation via stochastic dynamics, not heuristic feature aggregation. Also, we additionally conducted experiments to compare DW-MALA with the recent baselines that the reviewer mentioned. To promptly reflect this feedback, we have included the most recent paper (Bae et al., ICLR 2025), whose name is denoted as UHerding, and conducted the comparison on image classification on CIFAR-10, CIFAR-100, and Tiny-ImageNet. The results at each iterations are summarized in the tables below, which show that DW-MALA still outperforms the recent baseline, particularly in the early acquisition iterations.
> - Results on CIFAR-10
> | Method    | 1                        | 2                                        | 3                                        | 4                                        | 5                                        | 6                                        | 7                                        | 8                                        | 9                                        | 10                                       |
> |-----------|--------------------------|------------------------------------------|------------------------------------------|------------------------------------------|------------------------------------------|------------------------------------------|------------------------------------------|------------------------------------------|------------------------------------------|------------------------------------------|
> | UHerding  | $0.4431 \pm 0.0027$      | $0.6011 \pm 0.0017$               | $0.6480 \pm 0.0027$                      | $0.7106 \pm 0.0019$                      | $0.7277 \pm 0.0022$                      | $0.7528 \pm 0.0009$                      | $0.7694 \pm 0.0013$                      | $0.7762 \pm 0.0010$                      | $0.7854 \pm 0.0014$                      | $0.8036 \pm 0.0010$                      |
> | DW-MALA   | $0.4490 \pm 0.0014$      | **$0.6899 \pm 0.0016$**                  | **$0.7225 \pm 0.0009$**                  | **$0.7455 \pm 0.0021$**                  | **$0.7591 \pm 0.0019$**                  | **$0.7660 \pm 0.0013$**                  | $0.7751 \pm 0.0018$               | $0.7810 \pm 0.0010$                      | **$0.7951 \pm 0.0019$**                  | $0.8041 \pm 0.0026$               |
> - Results on CIFAR-100
> | Method    | 1                        | 2                        | 3                                  | 4                                  | 5                                  | 6                                  |
> |-----------|--------------------------|--------------------------|------------------------------------|------------------------------------|------------------------------------|------------------------------------|
> | UHerding  | $0.3499 \pm 0.0031$      | $0.3810 \pm 0.0009$ | $0.4066 \pm 0.0027$              | $0.4255 \pm 0.0008$                | $0.4356 \pm 0.0012$                | $0.4603 \pm 0.0030$                |
> | DW-MALA   | $0.3558 \pm 0.0030$      | **$0.4036 \pm 0.0026$**  | **$0.4292 \pm 0.0011$**           | **$0.4357 \pm 0.0022$**           | $0.4396 \pm 0.0025$        | **$0.4671 \pm 0.0009$**           |
>
> - Results on Tiny-ImageNet
> | Method    | 1                        | 2                        | 3                                      | 4                                      | 5                                      |
> |-----------|--------------------------|--------------------------|----------------------------------------|----------------------------------------|----------------------------------------|
> | UHerding  | $0.1113 \pm 0.0017$      | $0.2038 \pm 0.0023$ | $0.2605 \pm 0.0030$                  | $0.2913 \pm 0.0022$                    | $0.3088 \pm 0.0030$                    |
> | DW-MALA   | $0.1105 \pm 0.0029$      | **$0.2295 \pm 0.0012$**  | **$0.2682 \pm 0.0023$**                | **$0.2971 \pm 0.0028$**                | **$0.3115 \pm 0.0021$**                |

---

> ### Author Response · Authors · 2025-11-21
> **Response to Reviewer D34u (3)**
>
> 5. Experiments on ImageNet
>
> We additionally experimented with full ImageNet-1k classification with a linear-probe experiment using ResNet50 features. Due to high computational cost, we use fixed backbone features and trained additional two-layer perceptrons to evaluate our acquisition strategy. All the experimental settings followed those of Tiny-ImageNet described in Section 4.1.1. The results at each iteration are reported in the table below, which again show strong early-round gains of DW-MALA. We will add these results to the experiment section of the revised manuscript.
>
> | Method    | 1                        | 2                        | 3                        | 4                        | 5                        |
> |-----------|--------------------------|--------------------------|--------------------------|--------------------------|--------------------------|
> | Random    | $0.1485 \pm 0.0021$      | $0.2502 \pm 0.0023$      | $0.3414 \pm 0.0016$      | $0.4109 \pm 0.0029$      | $0.4532 \pm 0.0020$      |
> | Entropy   | $0.1472 \pm 0.0019$      | $0.2545 \pm 0.0034$      | $0.3510 \pm 0.0028$      | $0.4144 \pm 0.0030$      | $0.4571 \pm 0.0032$      |
> | Coreset   | $0.1480 \pm 0.0027$      | $0.2577 \pm 0.0030$      | $0.3569 \pm 0.0019$      | $0.4218 \pm 0.0024$      | $0.4592 \pm 0.0018$      |
> | VAAL      | $0.1475 \pm 0.0018$      | $0.2524 \pm 0.0026$      | $0.3462 \pm 0.0031$      | $0.4126 \pm 0.0027$      | $0.4551 \pm 0.0030$      |
> | LL4AL     | $0.1470 \pm 0.0024$      | $0.2540 \pm 0.0032$      | $0.3501 \pm 0.0033$      | $0.4156 \pm 0.0021$      | $0.4566 \pm 0.0023$      |
> | BADGE     | $0.1481 \pm 0.0017$      | $0.2515 \pm 0.0015$      | $0.3550 \pm 0.0022$      | $0.4236 \pm 0.0019$      | $0.4586 \pm 0.0028$      |
> | ProbCover | $0.1477 \pm 0.0020$      | $0.2579 \pm 0.0029$ | $0.3602 \pm 0.0025$ | $0.4311 \pm 0.0026$ | $0.4608 \pm 0.0034$ |
> | DW-MALA   | $0.1476 \pm 0.0031$      | $0.2803 \pm 0.0037$  | $0.3616 \pm 0.0018$  | $0.4307 \pm 0.0032$ | $0.4613 \pm 0.0015$ |
>
> 6. Discussion on $\alpha=0$ or $\alpha=1$
>
> Thank you for the thoughtful comment. We understand the concern. We would like to clarify that examining $\alpha$ only at the extreme settings ($\alpha$ = 0 or 1) does not fully reflect the intended behavior of our hybrid design. As shown in Figure 6, when $\alpha$ = 0 or $\alpha$ = 1, the method reduces to a single-criterion strategy, which results in noticeably worse performance, especially in the initial acquisition rounds where the balance between exploration and exploitation is most critical. Our motivation for DW-MALA is to simultaneously benefit from both the representativeness and diversity. The intermediate $\alpha$ values provide this desirable balance.
>
> 7. Gaussian Mixture Model as prior
>
> As the reviewer correctly mentioned, the Gaussian mixture in our implementation is intentionally used only as a prior, not as a fitted generative model. Therefore, it must not be trained or adapted to data.
>
> 8. Wall-clock time
>
> We sincerely apologize for the confusion caused. As the reviewer correctly pointed out, the reported values for Random and Entropy were mistakenly entered due to a rounding error during table formatting. We have re-measured and verified the wall-clock times on the same hardware environment (RTX A6000), and the corrected results are shown in the table below.
>
> | Method | Random | Entropy | Coreset | VAAL | LL4AL | BADGE | ProbCover | DW-MALA |
> |----------|--------|---------|---------|------|-------|-------|-----------|---------|
> | Time (sec)   |  0.0010   | 4.56    |  767.42    | 1003.16 |  938.88  |  947.05  |  472.61      | 884.90    |

---

> > ### Comment · Reviewer_D34u · 2025-11-27
> >
> > Thank you for the detailed rebuttal; many of my concerns are addressed. I still have a few comments regarding the discussion on heuristic metrics.
> >
> > > On “heuristic” representativeness metrics
> >
> > The authors argue that DW-MALA is more principled than clustering-based methods because it explicitly models the data distribution and selects points from high-density regions. However, I wouldn't frame clustering methods such as k-means clustering as merely heuristic. As shown in Bae et al. [1], MaxHerding is equivalent to greedy kernel k-means, which performs selection by maximizing the coverage of a kernel-induced data density e.g. RBF, defined over the penultimate feature space. In this sense, kernel k-means can be viewed as a principled estimator of density under its own modeling assumptions; much like DW-MALA relies on assumptions inherent to MALA-based density estimation. Therefore, while DW-MALA offers a compelling alternative, clustering-based representativeness metrics are not necessarily less principled; rather, they encode a different set of modeling assumptions.
> >
> > > Comparison with UHerding
> >
> > I appreciate the authors for providing additional comparison with UHerding on multiple datasets. For experiment setup, though, is it the same as that is specified in Section 4.1.1? I am asking because the numbers seem significantly lower than those reported in UHerding or relevant papers.
> >
> > > Experiments on ImageNet
> >
> > Here, Probcover seems not significantly better than random; it is quite different from what has been reported and I personally observed. Any insight for this?
> >
> > > Wall-clock time
> >
> > Thank you for the correction. I still do not understand the wall-clock time of some methods like coreset, LL4AL. Coreset does not require to construct a graph which is required in ProbCover; but coreset is significantly slower, why? Furthermore, LL4AL does not require any additional computation after one forward-pass unlike ProbCover or DW-MALA, but is much slower. Could you explain why this is the case?
> >
> > [1] Wonho Bae, Junhyug Noh, and Danica J Sutherland. “Generalized coverage for more robust low-budget active learning”, ECCV 2024.

---

> > > ### Author Response · Authors · 2025-12-02
> > > **Follow-up Responses to Reviewer D34u**
> > >
> > > We sincerely thank the reviewer for the insightful feedback. Below, we provide detailed responses to each point raised by the reviewer, along with additional experiments and analyses.
> > >
> > > 1. On “heuristic” representativeness metrics
> > >
> > > Thank you for this thoughtful clarification. We fully agree with the reviewer’s perspective, and we appreciate the opportunity to refine our framing. Indeed, kernel k-means and related clustering-based methods such as MaxHerding or Generalized Coverage are not merely heuristic, but are grounded in well-defined modeling assumptions (e.g., kernel-induced density notions). Our intention was not to diminish their theoretical grounding, but rather to convey the difference in modeling assumptions between these approaches and DW-MALA. In the revised manuscript, instead of calling previous methods “heuristic,” we will emphasize that DW-MALA is an alternative, density-centric modeling approach, not inherently “more principled.”
> > >
> > > 2. Comparison with UHerding
> > >
> > > We appreciate the reviewer for raising this concern. Our UHerding implementation followed the same setup as Section 4.1.1. To validate the correctness of the implementation, we re-ran UHerding following its original Low-Budget setting on TinyImageNet and compared its performance with our DW-MALA. The table below shows that the performance of UHerding is properly reproduced as in its original paper, with DW-MALA outperforming UHerding, especially in the early iterations.
> > >
> > > |Method|1|2|3|4|5|
> > > |-|-|-|-|-|-|
> > > |UHerding|0.0483|0.0752|0.0949|0.1077|0.1108|
> > > |DW-MALA|0.0484|0.0995|0.1035|0.1085|0.1114|
> > >
> > > 3. Experiments on ImageNet
> > >
> > > We agree that the gap between ProbCover and Random on our ImageNet-1k experiment is relatively small compared to the original ProbCover paper. We believe this behavior is mainly due to the specific experimental regime we adopted, which differs from prior work in several important aspects. In our setup, we use fixed ResNet-50 features and train only a small two-layer perceptron head. Under such a strong, pretrained feature extractor, even randomly selected labeled samples already form a reasonably good coverage of the feature manifold. As a result, the benefit of sophisticated coverage-based selection such as ProbCover is naturally reduced, because the backbone is no longer being adapted together with the acquisition process. Moreover, since the classifier on top of the frozen features is a shallow linear-like head, its capacity saturates relatively quickly as more labels are added. Once we reach that capacity limit, different acquisition strategies converge to similar final performance, which also contributes to the small gap at the final iteration. Despite these factors, ProbCover is still consistently slightly better than Random in our experiment, which is in line with its intended behavior, but the absolute gain is modest in this particular high-quality feature + large-batch setting. We will clarify this experimental regime in the revised manuscript and explicitly discuss why the benefit over Random is smaller than in the original ProbCover setup, which uses end-to-end training and different budget schedules.
> > >
> > >
> > > 4. Wall-clock Time
> > >
> > > We thank the reviewer for raising this concern. To verify the validity of our reported timings, we conducted an additional component-level profiling of all algorithms. For each method, we decomposed the total wall-clock time into substructure time. The table below shows the components of each algorithm with their wall-clock time reported, which explains our report on the wall-clock time in total.
> > >
> > > - Coreset
> > >
> > > |Component|Time|
> > > |-|-|
> > > |Pairwise Distance Matrix|46.38|
> > > |Optimization via K-Center Greedy|715.17|
> > >
> > > - LL4AL
> > >
> > > |Component|Time|
> > > |-|-|
> > > |Groundtruth-Loss Computation|19.85|
> > > |LPM Forward|3.52|
> > > |LPM Optimization|909.65|
> > >
> > > - ProbCover
> > >
> > > |Component|Time|
> > > |---|---|
> > > |Pairwise Probability Distance Matrix|10.21|
> > > |Coverage Computation|6.84|
> > > |Greedy Cover Selection Step|458.94|
> > >
> > > - DW-MALA
> > >
> > > |Component|Time|
> > > |---|---|
> > > |MALA Accept/Reject Phase |800.92|
> > > |KDE-based Representativeness Calculation|66.05|
> > > |Distance-based Diversity Calculation|12.67|
> > > |Score Calculation|3.74|

---

### Official Review · Reviewer_JXzN · 2025-11-02

**Soundness:** 3
**Presentation:** 3
**Contribution:** 3
**Rating:** 6
**Confidence:** 4

**Summary:**

The manuscript proposes a Langevin-based active sampling method that balances diversity and representativeness when querying data. Compared to deterministic approaches (e.g., DWDS), it introduces a stochastic mechanism intended to better handle multi-modal data distributions. The method is well-motivated, and experiments on image classification and detection show improvements over several baselines.

**Strengths:**

- Clear presentation of background and method.

- Thorough comparisons with many baselines; ablations on different representativeness measures and α-sensitivity indicate robustness.

- UMAP visualizations help illustrate the sampling behavior. (It would be helpful to add visualizations for more baselines and couple them with brief case studies, for instance, showcasing samples with high representativeness.)

- Runtime comparisons suggest DW-MALA is competitive with other non-trivial AL methods.

**Weaknesses:**

- The manuscript does not specify a tractable target density π, so it is unclear how ∇log π is computed/approximated in Eq. (11). This creates a derivation–implementation disconnect.

- Active learning performance can vary significantly across domains. The current evaluation is limited; extending to other tasks (e.g., NLP) would strengthen claims about robustness/generalizability.

- Presentation polish: Table 2 uses symbols (X / O / Δ) that are not explained; the corresponding discussion in the main text is also sparse.

- Minor: Since DWDS is a key baseline, including it directly in Fig. 2 would make trends easier to compare. Distinct line styles/markers (and a color-blind-friendly palette) could keep the figure readable even with more curves.

**Questions:**

- (W1) Without a tractable π, how is ∇log π obtained for Eq. (11)?

- Could the approach benefit from integrating deep neural architectures (e.g., Transformers) or diffusion models (given the sampling connection)? If so, what are the expected benefits and main challenges?

---

> ### Author Response · Authors · 2025-11-21
> **Responses to Reviewer JXzN (1)**
>
> We feel grateful for the constructive reviews. We provide responses to the reviews as well as supporting derivation below.
>
> 1. Computation of log-derivative
>
> We appreciate the reviewer for pointing out the need for clearer exposition. In the revised manuscript, we will explicitly clarify that $\nabla \log \pi (x)$ is computed from the KDE-based density estimator built from the MALA-generated sample set. Specifically, the KDE estimator allows the gradient to be computed analytically, where we provide the whole derivation below.
>
> We first define the kernel density estimator (KDE) and the Gaussian kernel as $  \hat\pi(x)
>     = \frac{1}{N} \sum_{i=1}^N K_h(x - x_i)$ and
> $ K_h(x - x_i)
>     = \frac{1}{(2\pi h^2)^{d/2}}
>        \exp\\left(
>          -\frac{\|x - x_i\|^2}{2h^2}
>        \right),$
> respectively; where $K_h(\cdot)$ is the $d$-dimensional Gaussian kernel with covariance
> matrix $h^2 I_d$ and $\|\cdot\|$ denotes the Euclidean norm.
>
> We next compute the gradient of the log-kernel with respect to $x$.
> To start with, taking logarithms of $K_h(x - x_i)$ gives
> $  \log K_h(x - x_i)
>     = \log\frac{1}{(2\pi h^2)^{d/2}}
>        - \frac{\|x - x_i\|^2}{2h^2}. $
>
> The first term is constant in $x$ and vanishes under differentiation.
> Hence, it holds that
> $  \nabla_x \log K_h(x - x_i)= -\frac{x - x_i}{h^2}.$
>
>
> Using the identity $\nabla_x K_h = K_h \,\nabla_x \log K_h$, we obtain $  \nabla_x K_h(x - x_i)= -\frac{1}{h^2}(x - x_i) K_h(x - x_i).$
>
> Hence, differentiating the KDE $\hat\pi(x)$ term-by-term yields
> $  \nabla_x \hat\pi(x) = -\frac{1}{Nh^2}\sum_{i=1}^N (x - x_i) K_h(x - x_i).$
>
>
> The gradient of the log-KDE is then obtained via the log-derivative rule,
>
> $  \nabla_x \log \hat\pi(x)
>     = \frac{\nabla_x \hat\pi(x)}{\hat\pi(x)} = -\frac{1}{h^2}
>        \frac{
>          \sum_{i=1}^N (x - x_i) K_h(x - x_i)
>        }{
>          \sum_{j=1}^N K_h(x - x_j)
>        } =-\frac{1}{h^2}
>        \sum_{i=1}^N
>          w_i(x) (x - x_i), $
>
> where we introduce normalized kernel weights $w_i(x)= \frac{K_h(x - x_i)}{\sum_{j=1}^N K_h(x - x_j)}$.
>
> Here, both the numerator and denominator are functions of $x$ through the
> kernel evaluations.
>
> We now show that the weighted sum
> $\sum_{i=1}^N w_i(x) (x - x_i)$ can be expressed in terms of
> a kernel-weighted mean of the samples as:
>
> $  \sum_{i=1}^N w_i(x) (x - x_i)
>     = \sum_{i=1}^N
>        \Bigl( w_i(x)x - w_i(x)x_i \Bigr)
>     = x \sum_{i=1}^N w_i(x)
>        - \sum_{i=1}^N w_i(x)x_i
>     = x - \sum_{i=1}^N w_i(x)x_i,$
>
> where we used the normalization $\sum_{i=1}^N w_i(x) = 1$.
>
> Substituting this identity back into the expression for
> $\nabla_x \log \hat\pi(x)$, we obtain two equivalent compact forms:
>
> $  \nabla_x \log \hat\pi(x)
>     = -\frac{1}{h^2}
>        \left(
>          x - \sum_{i=1}^N w_i(x)x_i
>        \right)
>     =
>        \frac{
>          \sum_{i=1}^N K_h(x - x_i) \nabla_x \log K_h(x - x_i)
>        }{
>          \sum_{i=1}^N K_h(x - x_i)
>        }.$
>
>
> Having said that, $\nabla \log \pi$ is not approximated heuristically but follows an analytic solution directly from the KDE representation. We will add this discussion to the revised manuscript.

---

> ### Author Response · Authors · 2025-11-21
> **Responses to Reviewer JXzN (2)**
>
> 2. Extending to NLP tasks
>
> We thank the reviewer for this valuable suggestion. We have conducted an additional experiment on a natural-language classification benchmark to better evaluate cross-domain robustness. Specifically, we used the 20 Newsgroups dataset for the classification task of predicting the topic given a document. We extracted feature representations using pretrained BERT embeddings as the input feature space for DW-MALA. We adopted VGG16 for our classification model, and used Adam optimizer with a learning rate of 1e-3. The initial labeled dataset consists of 100 documents, and we selected 100 additional ducoments for query at each iteration. We ran 5 independent trials, and report the results in the table below. The results show that DW-MALA consistently achieves the best or second-best performance across all acquisition rounds, while the advantage was the most significant in early iterations, demonstrating improved annotation efficiency when feature quality is still limited. We will include these results in the revised version of the manuscript as a new subsection in the experimental section, reinforcing that DW-MALA generalizes effectively beyond image domains.
>
> | Method    | 1                      | 2                                      | 3                                      | 4                                      | 5                                      | 6                                      | 7                                      | 8                                      | 9                                      | 10                                     |
> |-|-|-|-|-|-|-|-|-|-|-|
> | Random    | $0.3914 \pm 0.0026$    | $0.5469 \pm 0.0007$     | $0.5678 \pm 0.0015$                    | $0.6011 \pm 0.0013$                    | $0.5946 \pm 0.0030$                    | $0.6062 \pm 0.0028$                    | $0.6275 \pm 0.0035$                    | $0.6307 \pm 0.0009$                    | $0.6335 \pm 0.0020$                    | $0.6355 \pm 0.0007$                    |
> | Entropy   | $0.3913 \pm 0.0013$    | $0.5399 \pm 0.0022$                    | $0.5687 \pm 0.0007$      | $0.6023 \pm 0.0012$                    | $0.6152 \pm 0.0027$                    | $0.6254 \pm 0.0023$                    | $0.6301 \pm 0.0013$                    | $0.6327 \pm 0.0025$                    | $0.6387 \pm 0.0032$   | $0.6399 \pm 0.0006$      |
> | CoreSet   | $0.3912 \pm 0.0032$    | $0.5402 \pm 0.0028$                    | $0.5593 \pm 0.0017$                    | $0.6033 \pm 0.0011$                    | $0.6178 \pm 0.0037$      | $0.6204 \pm 0.0017$                    | $0.6276 \pm 0.0009$                    | $0.6314 \pm 0.0009$                    | $0.6347 \pm 0.0033$                    | $0.6367 \pm 0.0025$                    |
> | VAAL      | $0.3899 \pm 0.0032$    | $0.5401 \pm 0.0029$                    | $0.5640 \pm 0.0023$                    | $0.6028 \pm 0.0037$                    | $0.6165 \pm 0.0018$                    | $0.6229 \pm 0.0024$                    | $0.6289 \pm 0.0033$                    | $0.6321 \pm 0.0026$                    | $0.6367 \pm 0.0034$                    | $0.6387 \pm 0.0024$                    |
> | LL4AL     | $0.3901 \pm 0.0029$    | $0.5413 \pm 0.0007$                    | $0.5661 \pm 0.0013$                    | $0.6031 \pm 0.0015$                    | $0.6158 \pm 0.0009$                    | $0.6244 \pm 0.0013$                    | $0.6256 \pm 0.0009$                    | $0.6330 \pm 0.0015$                    | $0.6361 \pm 0.0026$                    | $0.6381 \pm 0.0018$                    |
> | BADGE     | $0.3905 \pm 0.0018$    | $0.5417 \pm 0.0013$                    | $0.5652 \pm 0.0015$                    | $0.6025 \pm 0.0036$                    | $0.6120 \pm 0.0027$                    | $0.6199 \pm 0.0025$                    | $0.6279 \pm 0.0011$                    | $0.6320 \pm 0.0029$                    | $0.6359 \pm 0.0011$                    | $0.6379 \pm 0.0018$                    |
> | ProbCover | $0.3918 \pm 0.0038$    | $0.5419 \pm 0.0026$                    | $0.5672 \pm 0.0024$                    | $0.6039 \pm 0.0028$             | $0.6172 \pm 0.0033$| $0.6266 \pm 0.0031$ | $0.6304 \pm 0.0013$  | $0.6345 \pm 0.0007$| $0.6372 \pm 0.0016$                    | $0.6395 \pm 0.0015$                    |
> | DW-MALA   | $0.3913 \pm 0.0013$    | $0.5602 \pm 0.0036$ | $0.5816 \pm 0.0034$    | $0.6043 \pm 0.0016$       | $0.6168 \pm 0.0027$                    | $0.6279 \pm 0.0019$    | $0.6309 \pm 0.0035$                | $0.6350 \pm 0.0021$    | $0.6379 \pm 0.0014$            | $0.6403 \pm 0.0014$  |

---

> ### Author Response · Authors · 2025-11-21
> **Responses to Reviewer JXzN (3)**
>
> 3. Symbols in Table 2
>
> We appreciate the reviewer’s feedback regarding the unclear use of the Δ symbol in Table 2. We acknowledge that the current notation may lead to confusion and will revise Table 2 accordingly. To clarify, AL-MALA directly queries the samples generated by the MALA sampler (denoted as $S_{mala}$), but these proposals are not necessarily located in high-density regions. The purpose of $S_{mala}$ is solely to be used for approximating the underlying distribution, so it does not guarantee representativeness of the selected instances. In contrast, DW-MALA first estimates density using $S_{mala}$, then explicitly selects top-K instances with the highest KDE-based representativeness score. Thus, AL-MALA does not incorporate representativeness in acquisition, and we will revise Table 2 to indicate X instead of Δ for representativeness. We will also add a legend describing each symbol for clarity.
>
> 4. Including DWDS in Figure 2
>
> We agree to the reviewer’s comment. In the revised manuscript, DWDS will be shown in Figure 2 across all datasets using distinct line styles.
>
> 5. Integrating deep neural architectures
>
> Using Transformer-based feature extractors would improve feature quality in early rounds, which directly benefits MALA’s density estimation. Also, our approach can naturally integrate more powerful density models such as diffusion models. In fact, the diffusion framework provides a learned score function $\nabla \log p(x)$, which can directly replace the KDE-derived log-gradient in Eq. (11). The posterior approximation and representativeness scoring steps up to Eq. (15) remain fully compatible with such integration. However, high computational overhead and training cost, particularly in early AL rounds where embeddings are still evolving, remain as challenges. We will add this discussion in the revised manuscript.

---

### Meta-Review · Area_Chair_pmxX · 2026-01-07

**Summary:**

This paper proposes a Langevin-based active sampling strategy that aims to balance diversity and representativeness in data querying. In contrast to deterministic methods such as DWDS, the proposed approach introduces a stochastic mechanism designed to better capture multimodal data distributions. The method is well motivated, and experimental results on image classification and object detection demonstrate improvements over several baseline methods.

The paper was reviewed by four reviewers, three of whom were inclined toward rejection. Reviewer D34u provided a detailed evaluation and engaged in extensive discussions with the authors, including aspects related to the code implementation. Despite these thorough exchanges, the overall review outcome suggests that the paper still has room for improvement.

**Reviewer Concerns:**

During the rebuttal phase, the authors addressed the following concerns:
- Details of the code implementation;
- The method derivation process;
- The scalability of the algorithm.

**Reviewer Scores:**

During the rebuttal phase, the authors responded to many of the reviewers’ questions, including an especially thorough discussion with Reviewer D34u. However, this also suggests that the paper still has several areas that require further improvement.

---

### Decision · Program_Chairs · 2026-01-26

Reject